# T cell assays differentiate clinical and subclinical SARS-CoV-2 infections from cross-reactive antiviral responses

Ane Ogbe [1,31], Barbara Kronsteiner[1,2,31], Donal T. Skelly [1,3,4,31], Matthew Pace[1,31], Anthony Brown [1,31], Emily Adland[1,31], Kareena Adair[5], Hossain Delowar Akhter[1], Mohammad Ali [1,2], Serat-E Ali[5], Adrienn Angyal[6], M. Azim Ansari[1], Carolina V. Arancibia-Cárcamo[4,7], Helen Brown[1], Senthil Chinnakannan [1], Christopher Conlon [2,3], Catherine de Lara[1], Thushan de Silva[6], Christina Dold[8,9], Tao Dong[10,11], Timothy Donnison [1], David Eyre[3,12], Amy Flaxman [13], Helen Fletcher [14], Joshua Gardner[5], James T. Grist[9,15,16], Carl-Philipp Hackstein[1], Kanoot Jaruthamsophon [5], Katie Jeffery [3], Teresa Lambe [13], Lian Lee [1], Wenqin Li[1], Nicholas Lim[1], Philippa C. Matthews [1,3], Alexander J. Mentzer [3,17], Shona C. Moore [18], Dean J. Naisbitt[5], Monday Ogese[5], Graham Ogg[3,9,10], Peter Openshaw [19], Munir Pirmohamed[5], Andrew J. Pollard [8,9], Narayan Ramamurthy[1], Patpong Rongkard[1,2,20], Sarah Rowland-Jones[6,21], Oliver Sampson [1], Gavin Screaton[17], Alessandro Sette[22,23], Lizzie Stafford[3], Craig Thompson [24], Paul J. Thomson[5], Ryan Thwaites [19], Vinicius Vieira [1,25], Daniela Weiskopf[22,23], Panagiota Zacharopoulou[1], Oxford Immunology Network Covid-19 Response T Cell Consortium*, Oxford Protective T Cell Immunology for COVID-19 (OPTIC) Clinical Team*, Lance Turtle [18,26,32], Paul Klenerman [1,3,7,9,32✉], Philip Goulder[25,32], John Frater [1,3,9,32], Eleanor Barnes [1,3,9,32] & Susanna Dunachie [1,2,3,9,20,32]

Identification of protective T cell responses against SARS-CoV-2 requires distinguishing people infected with SARS-CoV-2 from those with cross-reactive immunity to other coronaviruses. Here we show a range of T cell assays that differentially capture immune function to characterise SARS-CoV-2 responses. Strong ex vivo ELISpot and proliferation responses to multiple antigens (including M, NP and ORF3) are found in 168 PCR-confirmed SARS-CoV-2 infected volunteers, but are rare in 119 uninfected volunteers. Highly exposed seronegative healthcare workers with recent COVID-19-compatible illness show T cell response patterns characteristic of infection. By contrast, >90% of convalescent or unexposed people show proliferation and cellular lactate responses to spike subunits S1/S2, indicating pre-existing cross-reactive T cell populations. The detection of T cell responses to SARS-CoV-2 is therefore critically dependent on assay and antigen selection. Memory responses to specific non-spike proteins provide a method to distinguish recent infection from pre-existing immunity in exposed populations.

A full list of author affiliations appears at the end of the paper.

In late 2019, the new virus severe acute respiratory syndrome coronavirus 2 (SARS-CoV-2) emerged, causing the range of clinical diseases known as COVID-19[1,2]. While the majority of SARS-CoV-2 infections are either asymptomatic or result in mild disease, some individuals develop severe respiratory symptoms which may result in hospital admission and death leading to high global mortality[3,4], especially older adults and those with comorbidities[5]. Understanding the immune responses resulting from exposure to SARS-CoV-2 and distinguishing these from the responses made to seasonal coronaviruses, is a pre-requisite to defining immune correlates of infection and protection against subsequent SARS-CoV-2 disease. This in turn is centrally important in comparing with protective vaccine-induced immunity and may contribute to future public health policies including shielding advice.

Antibody responses to SARS-CoV-2 are important but remain complex. In a recent large-scale study of healthcare workers, PCR-confirmed SARS-CoV-2 infection resulted in measurable antibodies after 20 days in nearly all participants, with high specificity[6]. However, there is wide variability. Other studies have reported that antibodies may be absent early in the disease, levels of neutralising antibodies are highly variable[7], and antibody titres wane over time[8]. In contrast, studies of SARS-CoV infection indicate that T cell responses may be more durable[9]. A number of studies have demonstrated the presence of T cell responses to the virus during acute disease and in recovery. Using in silico-predicted HLA-class I and II peptide pools, CD4+ T cell responses to SARS-CoV-2 were demonstrated in all volunteers who had recovered from COVID-19 and CD8+ responses were demonstrated in 70%[10]. This study also found T cell reactivity to SARS-CoV-2 epitopes in 50% of archived samples from pre-pandemic (2015–2018) volunteers using a 24-h activation-induced markers (AIM) assay. Additionally, a Swedish study demonstrated a highly activated cytotoxic phenotype in acute disease and vigorous polyfunctional T cell responses in convalescent patients[11]. Interestingly, the latter study reported T cell responses to SARS-CoV-2 in seronegative household contacts, which may represent either infection without seroconversion or pre-existing cross-reactive immune memory to seasonal coronaviruses.

The role of prior exposure to human seasonal coronaviruses including alpha coronaviruses (HCoV-NL63 and HCoV-229E), and beta coronaviruses (HCoV-HKU1 and HCoV-OC43) as well as SARS-CoV and MERS-CoV, that may generate SARS-CoV-2 cross-reactive T cell immune responses, is of substantial interest. Whilst prior exposure to the original SARS-CoV and to MERS-CoV is rare and restricted to outbreaks, exposure to the seasonal human coronaviruses is widespread. Population sero-surveys have shown that detectable baseline levels of IgG against at least one of the four known HCoV is near universal[12–14], but there is evidence that re-infection with the same virus can occur[15,16]. T cell immunity to other coronaviruses is less well studied prior to the 2020 pandemic, but a recent study from Singapore demonstrated the presence of reactive responses to SARS-CoV-2 in people who had recovered from the SARS-CoV epidemic 17 years earlier, which are likely to represent cross-reactive memory[9]. Such cross-reactive responses to other CoV may be protective against SARS-CoV-2, be irrelevant, or could in theory contribute to immunopathology. The role of pre-existing cross-reactive T cell responses in immunity has been studied for other viruses including influenza and flaviviruses. In one study where such responses were fine-mapped, we observed that pre-existing cross-reactive responses to the dengue virus were linked to disease protection from Japanese Encephalitis, while symptomatic disease was linked to the emergence of strain-specific T cells[17].

Divergent data regarding SARS-CoV-2 T cell cross-reactivity have emerged so far: recent studies of T cell immunity to SARS-CoV-2 have reported levels of cross-reactive immunity to HCoV in SARS-CoV-2 unexposed populations of up to 50%[9–11,18–21] using a variety of immune assays. One such study from our centre[20] did not find significant ex vivo IFN-γ ELISpot responses to SARS-CoV-2 in uninfected, seronegative volunteers. The differences between these results might reflect the use of different assays employing a range of antigenic targets, peptide concentrations and proliferation times.

Here we set out to address two questions using a panel of T cell assays. First, do COVID-19 patients and seronegative controls show different levels of responsiveness in distinct assays of T cell function? Second, can T cell responses distinguish persons previously infected by SARS-CoV-2 from those previously infected by seasonal coronaviruses? We find—in a large cohort of people with a range of viral exposures—that cross-reactive memory responses to spike protein are almost universally detected using more sensitive assays, but that increasing viral exposure leads to an increase in magnitude and breadth of both effector and memory responses. These data have implications for our understanding of T cell cross-protection and for future studies of memory following the pandemic.

## Results

**Strong and broad IFN-γ ELISpot responses in convalescence.**
We first examined the T cell response to SARS-CoV-2 in freshly isolated peripheral blood mononuclear cells (PBMC) using an ex vivo IFN-γ ELISpot assay from 168 volunteers with PCR-confirmed SARS-CoV-2 infection, and 112 negative controls without evidence of SARS-CoV-2 infection (Supplementary Table 1). IgG antibody responses to spike measured by ELISA are shown in Fig. 1a and neutralising antibodies measured by a pseudoparticle assay are shown in Supplementary Fig. 1a. Firstly, we evaluated the magnitude of the T cell response to SARS-CoV-2 to assess the effector T cell response following stimulation of PBMCs with pools of overlapping peptides spanning all SARS-CoV-2 proteins except the non-structural ORF1 (Fig. 1b and Supplementary Table 2). We found responses to summed pools covering SARS-CoV-2 spike protein (12 minipools of 15-mers overlapping by 10 peptides referred to as P1–P12) (Fig. 1b, c), and the structural and accessory proteins (7 pools of 18-mers overlapping by 11 peptides covering E, M, NP, ORF3, ORF6, ORF7 and ORF8) (Fig. 1b, d). We also screened PBMCs with pools containing predicted optimal peptides targeting MHC Class II epitopes on the SARS-CoV-2 spike protein (CD4S), and other viral proteins (CD4All), and predicted Class I binding peptides split into CD8A and CD8B described in Grifoni et al.[13] (Fig. 1b and Supplementary Fig. 1c).

IFN-γ responses to spike (S) pools were seen in PBMC from 34/75 (45%) of convalescent volunteers tested (Fig. 1c) with high and frequent responses to some individual minipools including P2 (up to 313 SFC/10^6 PBMC) and P8 (up to 353 SFC/10^6 PBMC). We identified IFN-γ responses to the structural and accessory proteins in 65/103 (63%) of convalescent volunteers, with especially high-magnitude responses to the membrane (M) and nucleocapsid (NP) proteins (Fig. 1d). Combined, there was variation in the breadth and magnitude of SARS-CoV-2-specific responses (Supplementary Fig. 1b), and longitudinal follow-up studies underway will define the dynamics of the T cell response over time. IFN-γ responses were also seen in 24/29 (83%) of convalescent volunteers following stimulation with the four pools of predicted epitopes. Interestingly, we found especially high-frequency responses to the CD8A pool which comprises predicted

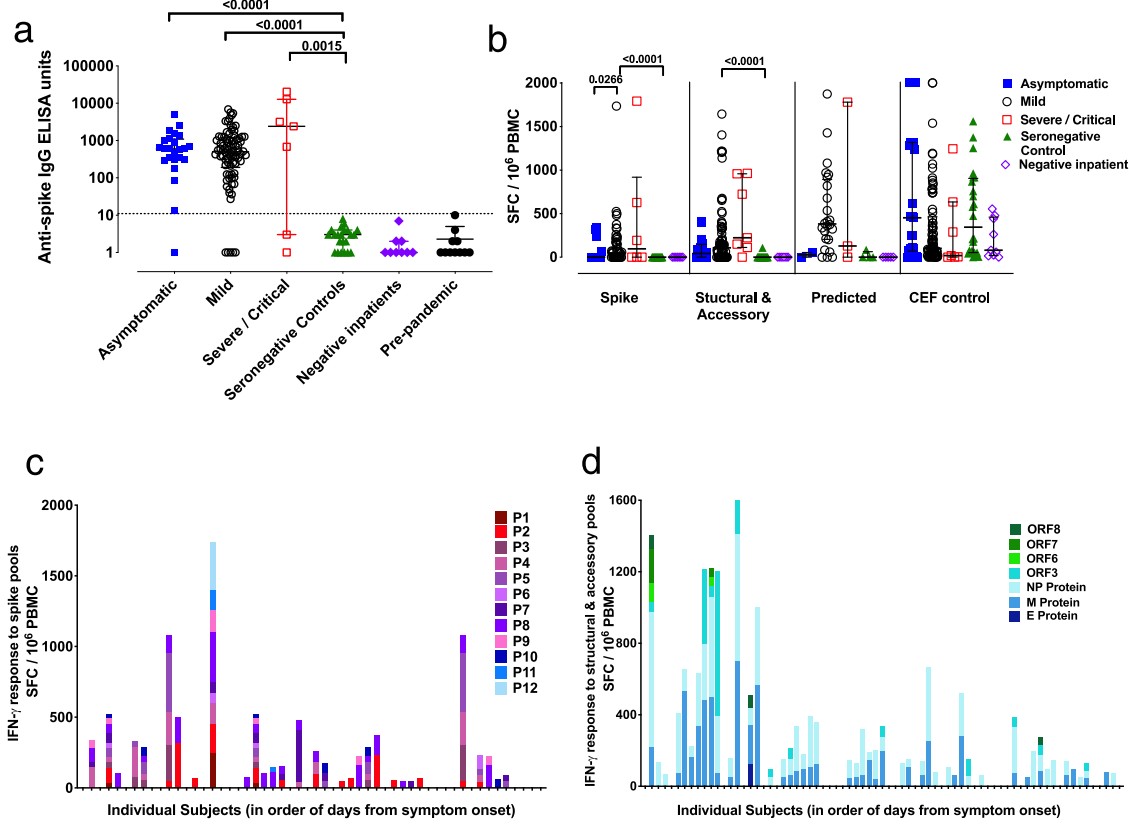

**Fig. 1 Magnitude and breadth of SARS-CoV-2-specific immune response. a** Total anti-SARS-CoV-2 spike IgG antibody titres by indirect ELISA[29] in 22 seronegative controls, 24 asymptomatic and 82 mildly symptomatic healthcare workers (HCWs) with PCR-confirmed SARS-CoV-2 infection, 7 hospitalised patients with severe or critical PCR-confirmed SARS-CoV-2 infection, 9 PCR-negative inpatient controls, and 11 pre-pandemic controls. **b** Ex vivo IFN-γ ELISpot showing the effector T cell responses to summed SARS-CoV-2 peptide pools spanning spike, accessory and structural proteins (E, M, NP, ORF 3, ORF6, ORF7 and ORF8), in silico-predicted pools[10] and the CEF T cell control panel in cohort groups as in **a**. **c** Ex vivo IFN-γ ELISpot showing the magnitude and breadth of effector T cell responses in 54 individual volunteers to 12 SARS-CoV-2 spike peptide pools (numbered P1 to P12) and **d** M, NP and accessory proteins ORF 3, ORF6, ORF7 and ORF8 in 73 HCWs convalescent with mildly symptomatic SARS-CoV-2 infection. X axis shows number of days from onset of symptoms (not to scale), with blank columns representing zero response in the individual tested at that time-point. SFC/10⁶ PBMC = spot-forming cells per million peripheral blood mononuclear cells, with background subtracted. Plots show median with error bars indicating ± IQR. Kruskal–Wallis one-way ANOVA, with Dunn's multiple comparisons test, was performed. Two-tailed P-values < 0.05 are shown on plots with Supplementary Table 3 showing full Kruskal–Wallis one-way ANOVA, with Dunn's multiple comparisons test for **b**. Source data are available in the source data file.

epitopes predominantly from the large ORF1[13] highlighting the need for further exploration of immune responses to this region (Supplementary Fig. 1c). Correlation analysis between the IFN-γ responses to spike peptide pools measured by ELISpot and anti-spike IgG measured by ELISA showed a significant positive correlation (Spearman's $R = 0.4587$; $P < 0.0001$) (Fig. 2a).

**IFN-γ responses to either M or NP correlate with total T cell responses**. There was a correlation between summed responses to spike and non-spike structural proteins (Spearman $R = 0.579$, $P < 0.0001$, Supplementary Fig. 1d), as well as the structural and accessory proteins and the predicted pools, indicating that when an individual mounted a T cell response to one part of the proteome they were likely to respond to another part, and responses declined with time from symptoms (Supplementary Fig. 1e, 1f). IFN-γ responses to either M or NP were correlates of the global response to spike, structural and accessory proteins (Fig. 2b, c), indicating that an assay to measure responses to M or NP could reflect the global effector T cell response.

We did not find a significant difference between IFN-γ ELISpot response and either age or sex (Supplementary Fig. 1g, h), but

larger studies including older adults are needed for further exploration.

**Sensitive proliferation assays demonstrate memory responses**. As our ELISpots assays were performed on total PBMCs, discrimination between distinct T cell lineages inducing the response was not possible. Moreover, the sensitivity of the ELISpot did not allow detection of responses in all COVID-19 recovered people. We, therefore, used a sensitive and functional flow cytometer-based assay capable of distinguishing the CD4+ and CD8+ T cell responses. For this, we used a T cell proliferation assay to gain further insights into the contribution and relative proficiency of the CD4+ or CD8+ T cell compartments to drive a proliferative anti-SARS-CoV-2 immune response in our convalescent HCW cohort. We first validated our assays on a small cohort of healthy control volunteers recruited for a hepatitis C virus (HCV) vaccine clinical trial pre-COVID19[22]. We showed that HCV seronegative control volunteers made strong proliferative responses to pools of optimal peptides covering Influenza, EBV, CMV and Tetanus (FEC-T) but as expected, not to peptides covering HCV NS3 or core proteins (Supplementary Fig. 2a–e). We then evaluated the ability of CD4+ and CD8+ T cells from the COVID-19

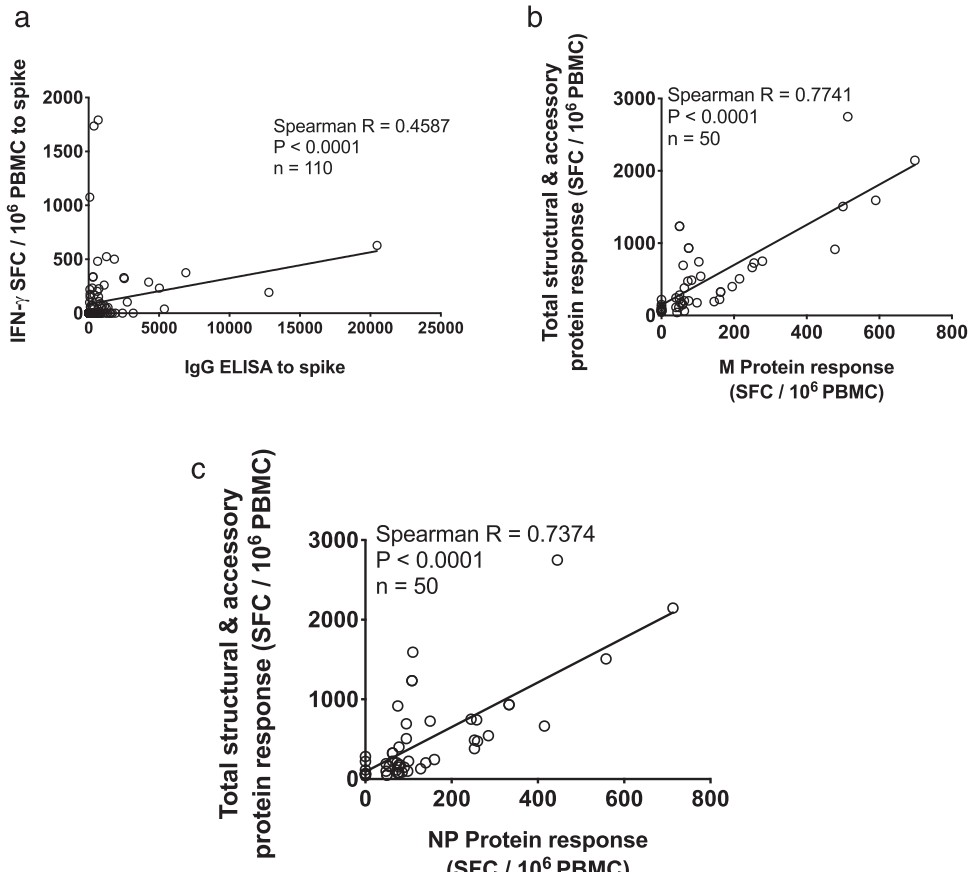

**Fig. 2 Correlation between antibody and total summed ex vivo ELISpot responses. a** Correlation between IgG ELISA to spike and ex vivo IFN-γ ELISpot summed response to spike ($n = 110$), the correlation between ex vivo IFN-γ ELISpot response to **b** M protein and **c** NP and total summed response to spike, E, M, N, ORF 3, ORF6, ORF7 and ORF8 ($n = 50$), SFC/$10^6$ PBMC = spot-forming cells per million peripheral blood mononuclear cells, with background subtracted. The correlation was performed via Spearman's rank correlation coefficient and comparison of two groups by two-tailed Mann–Whitney $U$ test.

convalescent HCW cohort to proliferate in response to peptide pools spanning key proteins from SARS-CoV-2. Live lymphocytes were separated into CD4$^+$ or CD8$^+$ T cells according to gating strategy and the frequency of proliferating cells analysed following a 7-day stimulation (Supplementary Fig. 2a). We found a high frequency of proliferating cells and broad targeting of SARS-CoV-2-specific CD4$^+$ and CD8$^+$ T cells (Fig. 3a, b, supplementary Tables 4 and 5) suggesting the establishment of a vigorous central memory population that may shape SARS-CoV-2 recall responses. The majority of people targeted T cell responses to M (69/107, 64% for CD4$^+$ and 50/107, 47% for CD8$^+$), NP (CD4$^+$ 63/107, 59%; CD8$^+$ 56/107, 52%) and ORF3 (CD4$^+$ 26/107, 24%; CD8$^+$ 24/107, 22%) and less frequently to ORF6 (CD4$^+$ 4/107, 4%; CD8$^+$ 2/107, 2%), ORF7 (CD4$^+$ 11/107, 10%; CD8$^+$ 6/107, 6%) and ORF8 (CD4$^+$ 13/91, 14%; CD8$^+$ 6/91, 7%) (Fig. 3c, d). This represents a higher sensitivity to detect antigen-specific T cell responses than in the ex vivo ELISpot assay. Although we observed a trend for the overall magnitude of the proliferating CD4$^+$ T cell response to SARS-CoV-2 peptide pools to be higher than that of the CD8$^+$ T cell-driven response, this did not reach significance for the peptide pools tested with the exception of M ($P = 0.0012$ by Mann–Whitney's $U$ test) (Supplementary Fig. 3a). Also of note, we did not find any difference in the magnitude of responding CD4$^+$ or CD8$^+$ T cells in individuals who had the asymptomatic disease (detected on HCW screening) compared with those who presented with mild symptoms (Supplementary Fig. 3b, c). Finally, the findings from

the proliferation assays were consistent with those generated by a second, shorter, assay measuring soluble lactate in supernatants obtained after only 4 days of stimulation, with SARS-CoV-2 convalescent volunteers showing strong M, NP and ORF 3-directed responses (Fig. 3e). Taken together, in our cohort of convalescent HCWs we show wide breadth and magnitude of T cell responses to SARS-CoV-2 proteins including both subunits of the spike (S1 and S2), and structural and accessory proteins.

**Specific CD4$^+$ and CD8$^+$ responses secrete multiple cytokines.** In order to determine the quality of the T cell response within our cohort, we used an intracellular staining (ICS) panel comprised of the activation marker CD154, degranulation marker CD107a and effector cytokines IFN-γ, TNF and IL-2 on freshly isolated PBMC. This allows for the assessment of both the contribution of CD4$^+$ and CD8$^+$ T cell responses as well as the pattern of cytokine response to SARS-CoV-2. As both ex vivo ELISpot and ICS assays measure effector memory cells, we first focused our ICS analysis on people who were ELISpot responders (>mean + 2 SD of the background) for M and NP pools, $n = 31$ and 41, respectively. Representative plots are shown in Supplementary Fig. 4. Levels of IFN-γ, IL-2 and TNF for these individuals are shown in Fig. 4a, c. For M pools, there was a larger CD4$^+$ T cell response compared to a CD8$^+$ response in terms of both IL-2 ($P < 0.0001$ by Wilcoxon's two-tailed test) and TNF ($P = 0.031$ by Wilcoxon's two-tailed test) (Fig. 4a). For NP pools, there was no

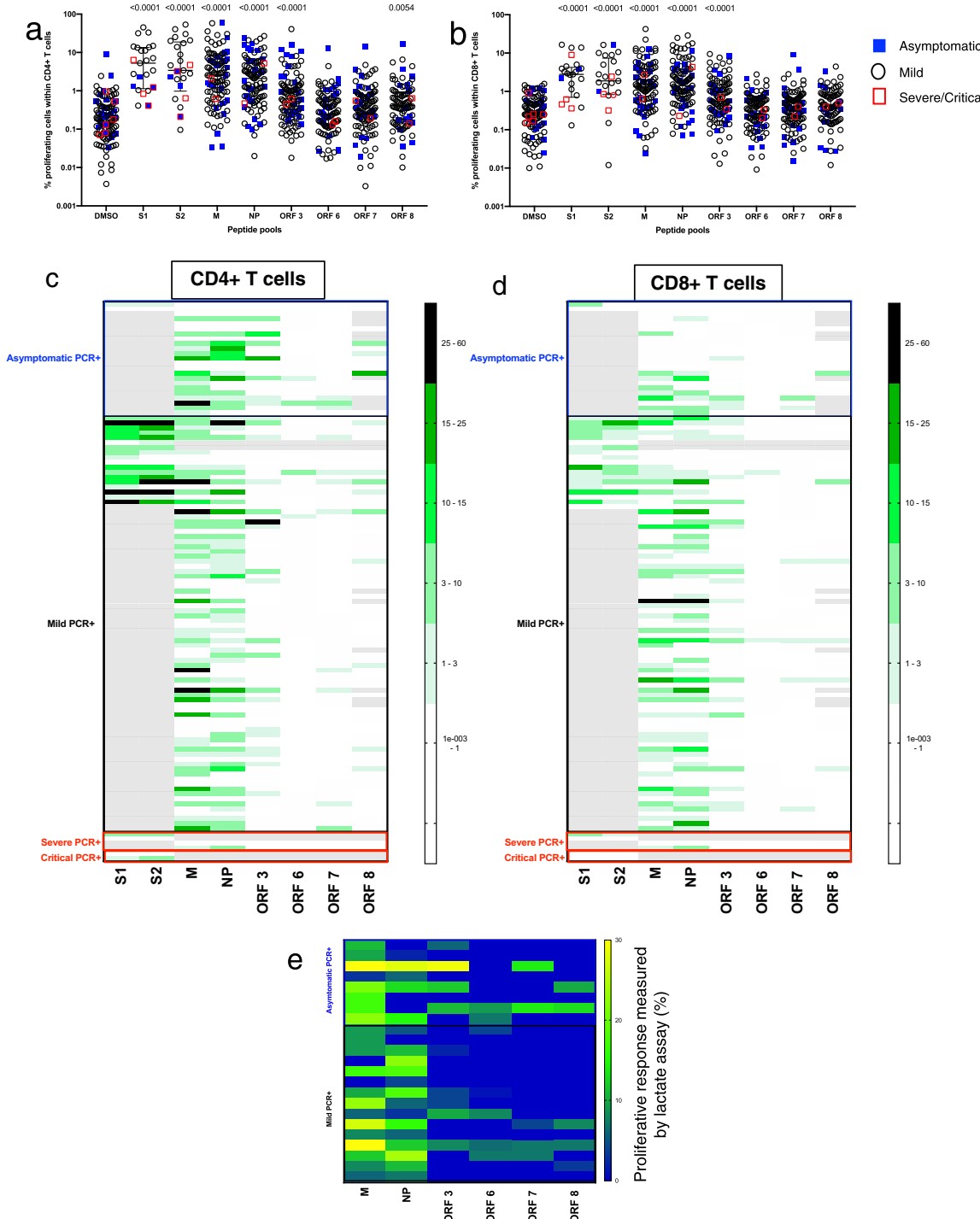

**Fig. 3 Proliferative responses in CD4$^+$ and CD8$^+$ T cells to key SARS-CoV-2 proteins.** Plot showing raw frequency (without background subtraction) of proliferating cells in response to peptide pool stimulation in 113 volunteers in **a** CD4$^+$ and **b** CD8$^+$ T cells to DMSO (media), and overlapping peptide pools spanning S1, S2, M, NP, ORF 3, ORF6, ORF7 and ORF8. **c** Heatmap showing the magnitude of proliferative responses to overlapping peptide pools spanning SARS-CoV-2 proteome in CD4$^+$ T cells and **d** CD8$^+$ T cells following background subtraction. Scales on the heatmap represent the magnitude of proliferating cells. Only data points >1% corresponding to mean + 2× SD in DMSO only well for both CD4$^+$ and CD8$^+$ T cells are shown. The grey box indicates absent data where tests were not run due to sample or peptide availability. **e** Cellular lactate proliferative response in convalescent mild and asymptomatic HCWs ($n = 23$ asymptomatic and mild symptoms) at day 4 revealed a variable response to M, NP, ORF 3, 6, 7 and 8. Heatmaps show background-subtracted responses. Each data point represents a single volunteer and plots show median with error bars indicating ± IQR. Where indicated, ns not significant, * = <0.05, ** = <0.01, *** = <0.001 and **** = <0.0001 by Kruskal–Wallis one-way ANOVA, with Dunn's multiple comparisons test for shown in Supplementary Tables 4 and 5. Number of volunteers for **a**–**d**: asymptomatic = 23, mild = 84, severe = 4, critical = 2.

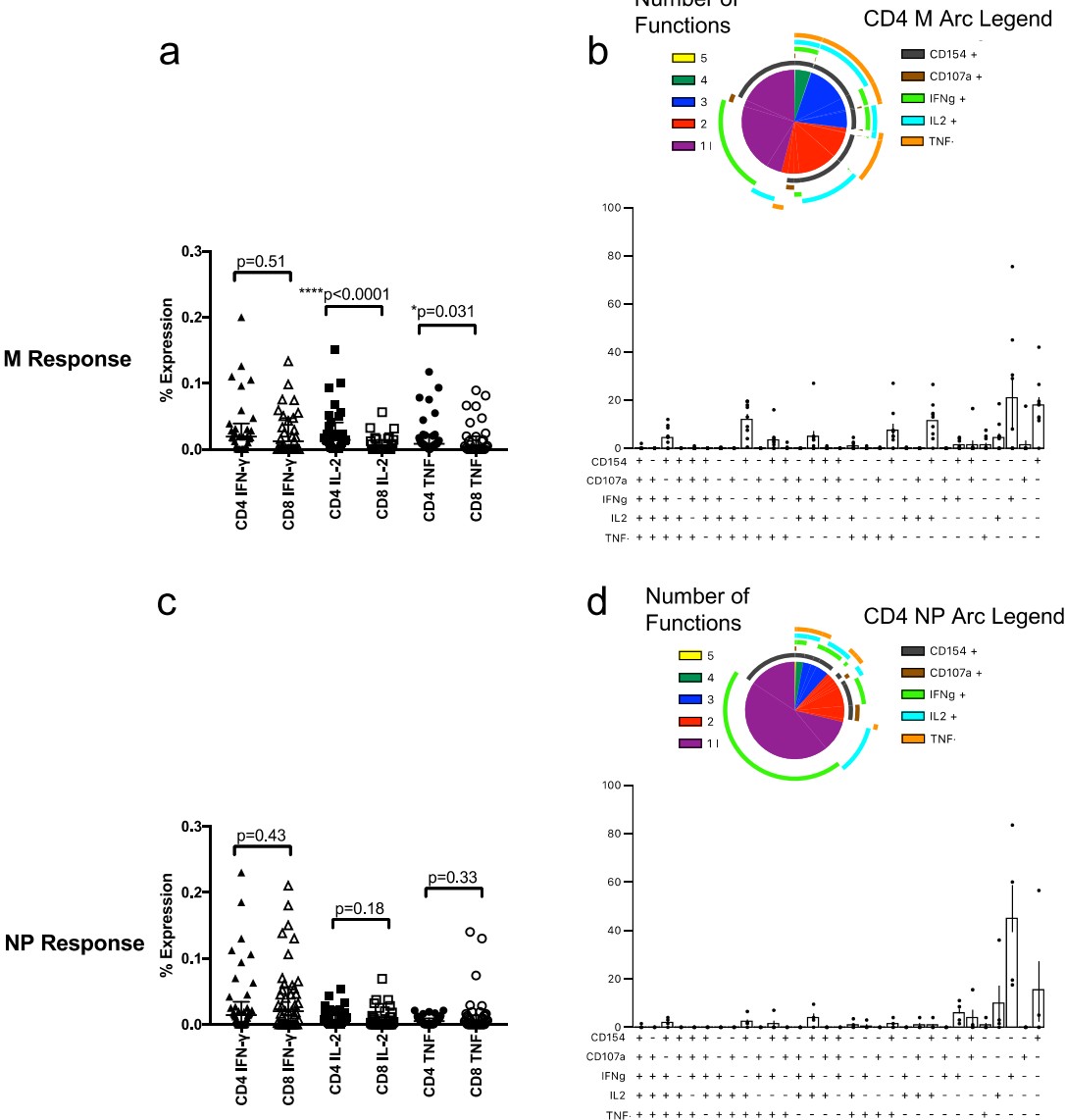

**Fig. 4 ICS responses in CD4+ and CD8+ T cells for M and NP pools in ELISpot positive individuals.** ICS was performed on individuals with convalescent mild cases and a positive ELISpot for the indicated peptides. PBMC were stimulated with 2 μg/ml peptide for 6 h. Expression levels of IFN-γ, IL-2 and TNF in CD4+ and CD8+ T cells using M pools are shown in **a**, $n = 31$. Bars represent median ± IQR. Statistics were performed using a two-tailed Wilcoxon matched-pairs signed-rank test between each cytokine in CD4+ vs CD8+ T cells. Boolean gates were then set and cytokine expression was examined in CD4+ T cells ($n = 9$) using SPICE (**b**). Error bars represent SEM for cytokine expression figures. Expression levels of cytokines using NP pools are shown in **c** ($n = 41$) with polyfunctionality analysis for CD4+ T cells (**d**) ($n = 4$) as above.

difference in the levels of IFN-γ, IL-2 or TNF expressed by CD4+ and CD8+ T cells (Fig. 4d). This difference between M and NP responses was not due to differences in the magnitude of the ELISpot response as they were statistically similar (median 85 vs 95 SFC/10^6 PBMC, $P = 0.37$ by Wilcoxon's two-tailed test). Neither was this difference due to patients with asymptomatic disease as there were similar numbers who were ELISpot positive for M ($n = 8$) and N ($n = 7$).

We then examined the number of functional markers co-expressed by these cells. After excluding individuals that did not have sufficient cells for multiple cytokine analysis either due to the low level of response (as median ELISpot level was <100/10^6 PBMC for both peptide pools) and/or the number of cells (particularly CD8+ T cells), we did not have a sufficient number of individuals with enough cells for CD8+ T analysis but did for

CD4+ T cells ($n = 9$ for M and $n = 4$ for NP). CD4+ T cells for both peptide pools expressed multiple cytokines with the majority of cells expressing one or two functional markers and up to four markers in CD4+ T cells (Fig. 4b, d).

We then performed ICS experiments using M, NP, S1 and S2 pools on an additional 26 SARS-CoV-2 PCR-positive individuals to compare the immune responses among these peptide pools (Supplementary Fig. 5). M, S1, and S2 pools all trended towards higher levels of IL-2 expression by CD4+ T cells compared to CD8+ T cells (Supplementary Fig. 5a, e, g, $P = 0.051$, $P = 0.055$, $P = 0.016$, respectively. All done by Wilcoxon's two-tailed test). Stimulation with M pools also resulted in significantly higher expression of IFN-γ by CD4+ T cells (Supplementary Fig. 5a, $P = 0.044$ by Wilcoxon's two-tailed test) while NP pools trended towards higher IFN-γ expression in CD8+ T cells

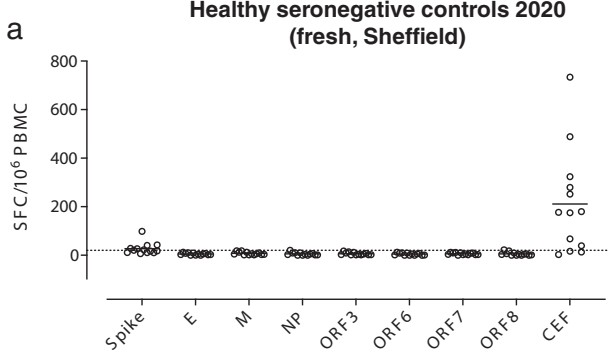

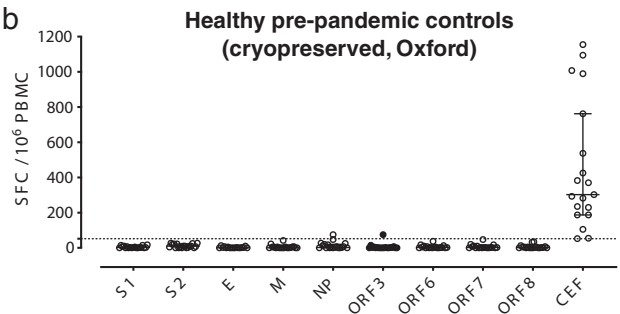

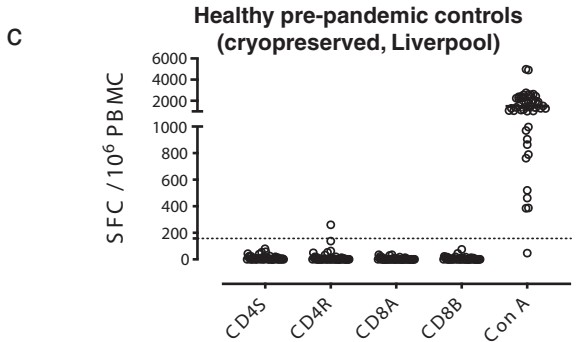

**Fig. 5 Ex vivo ELISpot responses in seronegative controls.** Ex vivo IFN-γ ELISpot responses to summed SARS-CoV-2 peptide pools spanning spike, accessory and structural proteins (E, M, NP, ORF 3, ORF6, ORF7 and ORF8) and CEF T cell control panel in **a** freshly isolated peripheral blood mononuclear cells (PBMC) from seronegative controls in Sheffield, UK ($n = 13$), and **b** cryopreserved PBMC from pre-pandemic healthy controls in Oxford, UK ($n = 19$). **c** Ex vivo IFN-γ ELISpot responses to in silico-predicted epitope pools[10] cryopreserved PBMC from pre-pandemic healthy controls in Liverpool, UK ($n = 48$). Responses are shown with background subtracted, a line represents mean +2 stand deviations of responses to the background.

(Supplementary Fig. 5d, $P = 0.066$ by Wilcoxon's two-tailed test). We again examined cytokine expression in these patients. The low level of responses, as these individuals were not screened using ELISpot, prevented us from effectively examining CD8+ T cell cytokine expression. However, there were sufficient individuals with enough responding cells to examine CD4+ T cytokine expression for M ($n = 10$), NP ($n = 4$), S1 ($n = 6$) and S2 ($n = 3$). The vast majority of cells expressed 1–2 functional markers (with similar patterns), with small populations of cells expressing 3 or 4 (Supplementary Fig. 5). These results were similar to the ELISpot positive individuals mentioned above.

**Uninfected show strong proliferative responses to spike**. We studied SARS-CoV-2 seronegative controls (Fig. 1a) for whom we also evaluated T cell responses to SARS-CoV-2 peptides using IFN-γ ELISpot, ICS and proliferation assay. In contrast to convalescent HCWs, SARS-CoV-2-specific IFN-γ responses were scarcely seen in any of the SARS-CoV-2 peptide pools as measured by ex vivo ELISpot assays in 23 seronegative healthy control volunteers (Fig. 1b). Responsiveness to common antigens (CEF-T) in these control volunteers indicated that there were no inherent defects in the ability of PBMCs from these donors to mount an antigen-driven immune response. This finding of a lack of response to SARS-CoV-2 peptides in seronegative control volunteers by an 18-h ex vivo IFN-γ ELISpot assay was confirmed in 13 volunteers by an independent laboratory in Sheffield, UK (Fig. 5a). We also evaluated cryopreserved PBMC from pre-pandemic healthy control archives and found minimal responses to spike, structural and accessory proteins in 19 volunteers in Oxford (Fig. 5b) and in the predicted epitope pools[10] in 48 people in Liverpool, UK (Fig. 5c).

However, using cellular proliferation assays on 20 seronegative volunteers, we show a high frequency of proliferating CD4+ and CD8+ T cells responding to the S1 and S2 subunit of the spike protein with a CD4+ T cell response detected in 17/20 (85%) and a CD8+ T cell response in 10/20 (50%) (Fig. 6a, b). In contrast, we observed weak or no CD4+ and CD8+ T cell proliferative responses to the structural and accessory proteins studied (M, NP, ORF3, ORF6, ORF7 and ORF8) (Fig. 6a, b). As the 20 seronegative participants were sampled in early 2020, we also analysed 15 cryopreserved samples from 2008 to 2019 (pre-UK COVID19 pandemic) to exclude the possibility of asymptomatic and undetected prior infection. Similar to the pandemic seronegative controls, we found no or low effector T cell responses by ELISpot assay to any of the spike, structural or accessory proteins (Fig. 5b), but as for the pandemic seronegative controls we detected robust T cell responses by proliferation assay to spike proteins S1 and S2, which was of greater breadth in the CD4+ T cells compared to their CD8+ T cell counterparts (Fig. 6c, d). The responses show a CD4+ skew with 15/15 showing a CD4+ T cell response and only 8/15 showing a CD8+ T cell response above background level. Most importantly, there was very limited cross-reactivity to the structural and accessory proteins as measured by the proliferation assay. As with the convalescent HCW cohort, we also performed a cellular lactate assay using supernatants obtained after 4 days of stimulation on 8 of these people. We confirm cross-reactive responses to spike S1 and S2 subunits, and non-existent or minimal responses in supernatants obtained from M, NP and accessory protein-stimulated PBMCs (Fig. 6e). We compared the magnitude of the proliferative responses to the different SARS-COV-2 peptide pools in seronegative controls from 2020, symptomatic and asymptomatic SARS-COV-2 PCR+ volunteers (Fig. 6f, g). We found no difference in the spike—S1 and S2—responses but the higher magnitude of proliferative responses to M and NP in both CD4+ and CD8+ T cells and ORF3 and ORF8 in CD8+ T cells alone in people who had tested positive to SARS-COV-2 (Fig. 6f, g). For confirmation, we also compared the magnitude of proliferative T cell responses in SARS-COV-2 seronegative controls from 2020 with the cryopreserved pre-pandemic seronegative controls and found the magnitude of proliferative cells in these two seronegative groups to be similar (Supplementary Fig. 6a, b). These results, in addition to our earlier results from people who did not generate effector T cell responses to spike peptides in the IFN-γ ELISpot assay (Figs. 1b and 5a, c), demonstrate the consistent evidence of proliferative T cell memory to spike protein in the pre-existing T cell repertoire of people naive to SARS-CoV-2.

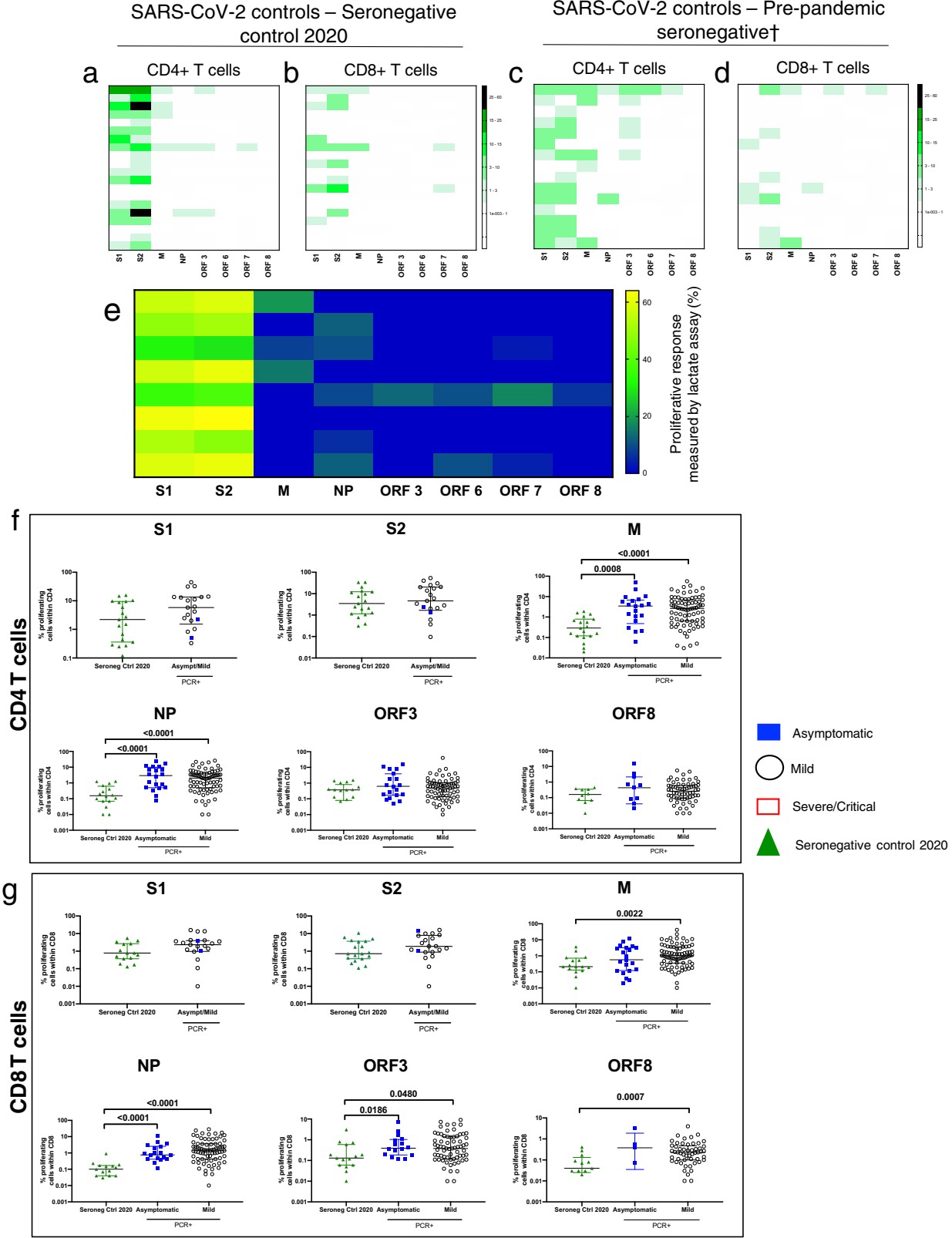

The robust T cell memory responses in the control groups to spike protein are clear in our data, and consistent with a recent study that assessed cross-reactivity using a different assay system[21] addressing the issue of whether this reflects prior exposure to HCoVs. To explore this further, we first tested our cohort using serologic assays and found universally high exposure to four circulating HCoVs, as also seen in other studies[9–11],[18–21] (Supplementary Fig. 7). We next addressed the homology between S and HCoVs at the level of 9-mer peptides and compared this to M and NP (Supplementary Fig. 8a) The median alignment score for each spike, NP and M proteins was 8.5, 7.5 and 6.5, respectively, which indicates that

**Fig. 6 Cross-reactive T cell response in seronegative controls from 2020 and pre-COVID19 pandemic. a** Heatmaps showing CD4$^+$ and **b** CD8$^+$ T cell proliferative responses in fresh PBMCs from healthy seronegative controls ($n = 20$). **c** heatmaps showing the magnitude of cross-reactive responses in CD4$^+$ and **d** CD8$^+$ T cell response in cryopreserved samples obtained pre-COVID19 pandemic ($n = 15$). Only data points >1% corresponding to mean + 2× SD in DMSO only well for both CD4$^+$ and CD8$^+$ T cells are shown in heatmaps. **e** Heatmap measuring the lactate proliferative response in both healthy seronegative controls at day 4 revealed a strong response to spike (all S1 and S2 values divided by 2 for ease of viewing) as well a small, variable, response to SARS-CoV-2 peptide pools. **f** comparative analysis of peptide pool-specific proliferative response to SARS-CoV-2 proteins in CD4$^+$ and **g** CD8$^+$ T cells in SARS-CoV-2 seronegative controls during COVID pandemic and PCR$^+$ volunteers. All data plotted are background subtracted (number of volunteers: seronegative control 2020 = 20, pre-pandemic seronegative = 15). For statistical comparison, all data points have been included for all groups. Each data point represents a single volunteer and plots show median with error bars indicating ± IQR. Comparison of two groups was done by two-tailed Mann–Whitney $U$ test.

on average SARS-CoV-2 spike protein peptides are more conserved relative to the HCoVs than M and NP proteins (Mann–Whitney $U$ test $P$-values: S vs $N < 2.2 \times 10^{-16}$, S vs $M < 2.2 \times 10^{-16}$). Additionally, focusing on the top 5% of most homologous peptides (alignment score > 18.5) across the three proteins, we observed that 61%, 20% and 19% of the peptides were in S, NP and M proteins, respectively. Finally, since direct cross-reactivity has been shown at the level of a key set of 61 SARS-Cov-2-derived peptides[21], we assessed to what extent this dataset supported enhanced reactivity to these peptides derived from S vs non-S antigens. We observed strong skewing towards S ($P < 0.0001$ by two-tailed Welch's ANOVA), with such peptides distributed in both S1 and S2, as seen in this study (Supplementary Fig. 8b, c). Overall, these data suggest that S contains a pool of T cell epitopes with evident conservation and cross-reactivity and in independent analyses of similarly HCoV-exposed populations are associated with common T cell reactivity to S.

**T cell responses in seronegative exposed healthcare workers.** Finally, to explore the use of these T cells assays to identify people potentially exposed to SARS-CoV-2, we recruited a group of 10 highly exposed healthcare workers working in acute medicine who had experienced symptoms compatible with COVID-19 but had not received PCR testing at the time of symptoms, or tested negative, and were subsequently seronegative (Supplementary Table 6). 3/10 of these healthcare workers showed effector T cell responses by ELISpot assay to S1, S2, M or NP (Fig. 7a) whilst 8/8 of those tested showed M and/or NP-specific T cell responses in the proliferation assay compatible with prior infection (Fig. 7b, c). Analysis of the breadth of SARS-CoV-2 antigen targeted by the responding CD4$^+$ and CD8$^+$ T cells shows that in the highly exposed doctors the CD4$^+$ and CD8$^+$ T cell response is directed to a broader number of structural (M and NP) and accessory (ORFs 3, 6, 7 and 8) SARS-CoV-2 peptide pools (Fig. 7d, e). This reached statistical significance for both CD4$^+$ and CD8$^+$ T cells compared to seronegative control groups (Fig. 7d, e). Lastly, we compared the magnitude of the T cell response to SARS-CoV-2 structural and accessory proteins in the three groups – the highly exposed HCWs, seronegative controls from 2020 and pre-pandemic seronegative controls (combined into one group). We found a significantly higher magnitude of CD4$^+$ but not CD8+ T cells proliferating in response to the M, N, ORF3, 6, 7 and 8 in the highly exposed doctors (Fig. 7f, g).

**Discussion**
As the global COVID-19 pandemic continues, it is important to define which immune responses are important for protection. In this study, we have used distinct T cell assay platforms across the same individuals to characterise the differences between T cell responses associated with recent SARS-CoV-2 infection and long-term cross-reactive memory T cell responses in unexposed populations. The effector T cell response as measured by our 18-h

ex vivo IFN-γ ELISpot assay showed a remarkable absence of SARS-CoV-2-specific responses in most of the healthy seronegative volunteers. The ELISpot assay, therefore, represents potential as a suitable assay platform for the identification of recent infection with SARS-CoV-2. However, this ELISpot assay did not detect SARS-CoV-2-specific T cell responses in all people with recent PCR-confirmed infection, and the longevity of such responses requires further analysis in longitudinal studies. We already noted a significant inverse correlation with magnitude by ELISpot assay over time in the short follow-up performed here, and ongoing work with the current convalescent HCW cohort will define the durability of these T cell responses induced by SARS-CoV-2 infection.

In contrast, the same healthy people showed responses to the S1 and S2 subunits of spike protein in a 7-day CTV proliferation assay, confirmed by analysis of lactate production. The most likely explanation for this is that people developed cross-reactive memory responses to the spike protein of seasonal coronaviruses that circulate in the UK (and to which they all exhibit serologic responses), although cross-reactivity from other human microorganisms is also possible, as has been described for HIV, influenza and Ebola epitopes in naive volunteers[23,24]. The relative focus of such pre-existing responses on S1 and S2 is consistent with recently-published data[21] where proof-of-principle for cross-reactivity with HCoVs was shown. It is also supported by our informatic studies showing enrichment for peptides conserved across such coronaviruses in S compared to M and NP antigens. Thus although cross-reactive, pre-existing T cell responses can occur throughout the genome, we hypothesise that the relative conservation and also the large size of the spike may lead to an accumulation of the consistent S-specific memory responses in the control groups seen using our assays.

Individuals in convalescence from SARS-CoV-2 infection showed strong and broad effector CD4$^+$ and CD8$^+$ T cell responses to peptides spanning the SARS-CoV-2 genome as previously reported, with CD4$^+$ T cells showing a polyfunctional response[20]. Unfortunately, there was an insufficient number of individuals with sufficient cell numbers available to effectively examine CD8$^+$ T cell cytokine expression. ELISpot responses to the M and NP proteins were especially frequent and high, and each correlated with the summed response to spike, structural and accessory peptides, indicating their suitability as antigens for screening individuals and populations for evidence of T cell immunity following exposure to SARS-CoV-2. Additionally, memory responses to M and NP were frequent and strong in the proliferation assay for people in convalescence from SARS-CoV-2 but significantly less so in the seronegative control volunteers, further supporting the use of these antigens as markers of T cell responsiveness more closely linked with SARS-CoV-2 exposure. Further mapping studies could identify peptides with the highest sensitivity and specificity for SARS-CoV-2 infection, with potential for use in defining T cell immunity at an individual and population level.

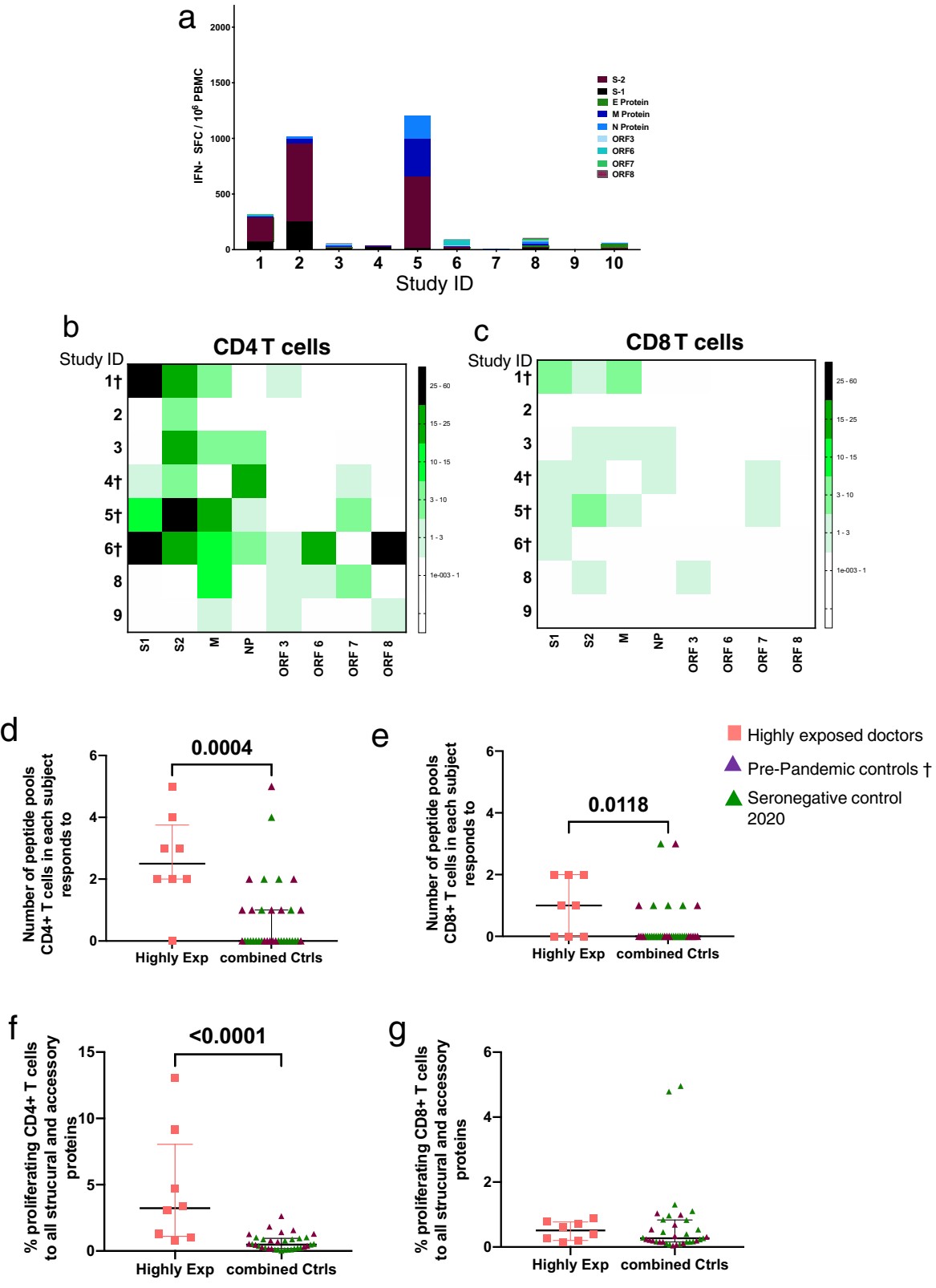

The existence of substantial T cell cross-reactivity to SARS-CoV-2 from prior HCoV exposure has been demonstrated in non-SARS-CoV-2 infected populations from a range of geographical locations[9–11,18,19,21]. Here, we demonstrate the use of the ELISpot assay to identify SARS-CoV-2-specific responses, and our finding of absent T cell responses in unexposed volunteers

was confirmed by similar results in our three independent laboratories (Universities of Oxford, Liverpool and Sheffield). T cell assays vary in their sensitivity, influenced by cell number, incubation time, antigen choice and concentration and markers of T cell activity measured. Our ELISpot assay does not detect the T cell responses in unexposed populations to spike and other

**Fig. 7 T cell response in highly exposed seronegative controls. a** Ex vivo IFN-γ ELISpot responses to summed SARS-CoV-2 peptide pools spanning spike, accessory and structural proteins (E, M, NP, ORF 3, ORF6, ORF7 and ORF8) in highly exposed HCWs working in acute medical care who experienced a COVID-19-compatible illness without PCR testing and were subsequently seronegative. Responses are shown with background subtracted, $n = 10$. **b** Heatmaps showing CD4$^+$ and **c** CD8$^+$ T cell proliferative responses in the same population of highly exposed HCWs. All data plotted are background subtracted, $n = 8$ (cells unavailable for 2). **d** Breadth of responses to structural and accessory proteins from SARS-COV-2 in CD4$^+$ and **e** CD8$^+$ proliferating T cells. **f** Magnitude of responding CD4$^+$ and **g** CD8$^+$ T cells to structural and accessory proteins from SARS-CoV-2 (M, NP, ORF3, 6, 7, 8). Each data point represents a single volunteer and plots show median with error bars indicating ± IQR. Comparison of two groups was done by two-tailed Mann–Whitney $U$ test. Study ID with † was assessed from cryopreserved samples. Proliferation assay for individuals 7 and 10 was not performed. $N = 35$ for combined control.

SARS-CoV-2 proteins reported elsewhere. This may be due to the relatively low cell number used in our assay (200,000 per well) but most likely the focus on IFN-γ release rather than detection of cell activation markers.

Most convalescent people in the study made antibodies, as detected by IgG ELISA and pseudoparticle neutralisation assay. Emerging literature suggests that SARS-CoV-2 IgG titres meeting the threshold for positivity may be relatively short-lived[8,25]. The current study represents a cross-sectional 'snapshot' in time of human T cell responses to SARS-CoV-2 after infection. Ongoing follow-up studies of this cohort and surveillance[26] for re-infection aligned to the UK SIREN study[27] will allow further delineation of the time course of T cell responses in parallel with humoral responses, and the timing of any assay must be taken into account in defining its utility. While an association is seen between antibody and ELISpot in the PCR-positive cohort, a disjunct exists between the antibodies and memory responses, since strong spike responses can be seen in the PCR-negative/unexposed and pre-pandemic groups. We need to assess in future whether any relationship exists between the levels of these responses and levels of seroreactivity to HCoVs.

Our study of the large ORF1 was restricted to the use of the in silico-predicted pool CD8A[10], where a high magnitude of responses was seen. Further work will characterise the time course of T cell responses observed in this cohort, evaluate the ability of our assays to correctly distinguish individuals with confirmed SARS-CoV-2 infection from unexposed controls, and prospectively seek to identify the relationship between measurable T cell immunity to the SARS-CoV-2 proteome and subsequent primary or secondary infection with SARS-CoV-2.

Overall, we have shown that assessments of T cell immunity using different assays but with the same antigens give very different results. Our ELISpot measure of ex vivo IFN-γ release is valuable in defining the potential role of T cell immunity in recently infected donors without cross-reactivity in unexposed people. In contrast, our proliferation assay allows dissecting out pre-existing vs SARS-CoV-2 induced immune responses by examining responses to different antigens. Our proliferation assay demonstrates widespread T cell memory responses to spike in both SARS-CoV-2 infected and unexposed people, whilst T cell memory responses to M and NP are more characteristic of previous SARS-CoV-2 infection. These two assays, in combination with the panel of antigens, can now allow us to address critical questions about dissecting the role of T cells—induced by SARS-CoV-2, and possibly HCoVs and other antigens—in immune protection in the future.

## Methods

### Ethics statement.
Human study protocols were approved by the research ethics committee (REC) at Yorkshire & The Humber— Sheffield (GI Biobank Study 16/YH/0247). The study was conducted in compliance with all relevant ethical regulations for work with human participants, and according to the principles of the Declaration of Helsinki (2008) and the International Conference on Harmonization (ICH) Good Clinical Practice (GCP) guidelines. Written informed consent was obtained for all patients enrolled in the study.

### Study volunteers.
SARS-CoV-2 positive individuals: Healthcare workers at Oxford University Hospitals NHS Foundation Trust who tested positive for SARS-CoV-2 following either presentation to the hospital's Occupational Health Department with symptoms or having a positive PCR test on the staff screening programme[26] were asked to indicate whether they were willing to be contacted by researchers. Individuals who agreed to be contacted received an email invitation to participate in the study. Volunteers recruited from the staff screening programme were classified as asymptomatic if they did not report any symptoms of COVID-19 (including fever, shortness of breath, cough, loss of taste or smell, sore throat, coryza or diarrhoea), either prior to staff screening or in the seven days following testing positive. In total 126 symptomatic and 33 asymptomatic people were recruited for this study. In addition, 9 hospitalised PCR-positive patients with WHO severe or critical COVID-19 were studied.

SARS-CoV-2 negative individuals (healthy controls): 30 healthy control volunteers in Oxford and 13 in Sheffield with no history of COVID-19 symptoms and no antibodies to SARS-CoV-2 spike protein detected by IgG ELISA were recruited. In addition, archived samples from 19 healthy control volunteers in Oxford who donated blood in the pre-pandemic period (2008–2019) were studied, alongside 48 healthy control volunteers from the pre-pandemic period in Liverpool. In addition, 9 hospitalised PCR-negative patients with other medical conditions were studied.

Highly exposed seronegative individuals (highly exposed HCWs): 10 acute medicine doctors, who worked in patient-facing services during the pandemic and experienced symptoms compatible with COVID-19, but did not receive PCR testing at the time of symptoms or tested negative, and were anti-spike IgG negative two months after the pandemic peak, were recruited as highly exposed seronegative HCW participants.

### Peripheral blood mononuclear cells.
Peripheral blood mononuclear cells were isolated by density gradient centrifugation using Lymphoprep$^{TM}$ (1.077 g/ml, Stem Cell Technologies)[28]. Plasma was collected and spun at $2000 \times g$ for 10 min to remove platelets before freezing at −80 °C for later use. PBMC were collected and washed twice with pre-warmed R10 media: RPMI 1640 (Sigma, St. Louis, MO, USA) supplemented with 10% heat-inactivated FCS (Sigma), 1 mM Pen/Strep and 2mM L-Glutamine (both from Sigma). After the second centrifugation, cells were resuspended in R10 and counted using the Guava® ViaCount$^{TM}$ assay on the Muse Cell Analyzer (Luminex Cooperation). The majority of assays were performed on freshly isolated PBMC during the first peak of the pandemic using available resources, and it was not possible to test all samples with all antigens. Assays performed on frozen samples are indicated in the manuscript.

### Antigens.
For functional assays, PBMC were stimulated with three groups of peptide pool for SARS-CoV-2: (1) Spike: 15-mers overlapping by 10 amino acid residues for spike (S), divided into 12 'minipools' P1–P12 (Proimmune)[29], and grouped into pools S1 (P1-6) and S2 (P7-12) for some assays (2) Structural and accessory proteins: 12-20-mer peptides overlapping by 10 amino acid residues for membrane protein (M), nucleoprotein (NP), envelope (E) protein, open reading frame (ORF) 3, 6, 7 and 8 (Proimmune)[20] and (3) Predicted epitope pools: predicted CD4$^+$ and CD8$^+$ pools[10] from the Sette laboratory, La Jolla Institute, CA, all used at a final concentration of 1–2 μg/ml per peptide depending on the assay. Lyophilised peptides were reconstituted in DMSO (Sigma).

### IFN-γ ELISpot assay.
The kinetics and magnitude of the cellular responses to SARS-CoV-2 were assessed by ex vivo IFN-γ ELISpot[28]. Fresh PBMC were used in all ELISpot assays unless otherwise indicated in figure legends. Briefly, 96-well Multiscreen-I plates (Millipore, UK) were coated for 3 h with 10 μg/ml GZ-4 anti-human IFN-γ (Mabtech, AB, Sweden) at room temperature. Fresh PBMC were added in duplicate wells at $2 \times 10^5$ cells in 50 μl per well and stimulated with 50 μl of SARS-CoV-2 peptide pools (2 μg/ml per peptide) as indicated in the figure legends and controls. R10 with DMSO (final concentration 0.4%, Sigma) was used as negative control and the following reagents were used as positive controls: CEFT peptide pool (2 μg/ml, Proimmune) and Concanavalin A (5 μg/ml final concentration, Sigma). After 16–18 h at 37 °C, 5% $CO_2$, 95% humidity, cells were removed and secreted IFN-γ was detected by adding 1 μg/ml anti-IFN-γ biotinylated mAb (7-B6-1-biotin, Mabtech) for 2–3 h, followed by 1 μg/ml streptavidin

alkaline phosphatase for 1–2 h (SP-3020, Vector Labs). The plates were developed using BCIP/NBT substrate (Pierce) according to the manufacturer's instructions. ELISpot plates were scanned on an AID ELISpot Reader (v.4.0) using the following settings: intensity min 12, size min 22, gradient min 4. Results were reported as spot-forming units (SFU) per million PBMC. The unspecific background (mean SFU from negative control wells) was always less than 50 SFU/$10^6$ PBMC and subtracted from experimental readings.

**Intracellular cytokine stimulation assay.** PBMC resuspended in R10 were plated at $1 \times 10^6$ live cells/well into 96-well round-bottom plates and stimulated with SARS-CoV-2 peptide pools (2 µg/ml per peptide) as indicated in the figure legends. Media containing DMSO (0.1%, Sigma) was used as negative control and PMA (0.05 µg/ml) with ionomycin (0.5 µg/ml, Sigma) as a positive control. CD107a BV421 (BD Biosciences), monensin (Biolegend) and Brefeldin A (MP Biomedicals) were added to cultures at a final concentration of 0.04 µg/ml, 0.16 µM, and 10 µg/ml, respectively, and cells were incubated for 6 h at 37 °C, 5% $CO_2$, 95% humidity. Plates were placed at 4 °C overnight and subjected to flow cytometry staining as described below. In addition to the three cytokines, CD107a was examined as was CD154 in $CD4^+$ T cells.

**Proliferation assay.** PBMCs from freshly isolated blood samples or cryopreserved samples (denoted with †) were twice washed with 1× PBS and stained using CellTrace® Violet (CTV, Life Technologies) at a final concentration of 2.5 µM for 10 min at room temperature. The reaction was quenched by adding cold FBS. CTV-labelled PBMC in RPMI containing 10% human AB serum (Sigma), 1 mM Pen/Strep and 2mM L-Glut were plated in 48 or 96-well round-bottom plates at 500,000 and 250,000 cells, respectively, and stimulated with peptide pools from SARS-CoV-2, FEC-T, HCV NS3 or HCV core protein (1 µg/ml per peptide). Media containing 0.1% DMSO (Sigma) representing DMSO content in peptide pools were used as negative control and 2 µg/ml phytohemagglutinin L (PHA-L, Sigma) as used as a positive control. Cells were subsequently incubated at 37 °C, 5% $CO_2$, 95% humidity for 5 days without media change or 7 days with media change on day 4 if cultures were kept beyond 5 days. At the end of incubation, cells were subjected to flow cytometry staining as described below. Responses above 1% were considered true positive. To determine the breadth of antigenic response targeted by T cells, the number of peptide pools that each volunteer responded to was counted. To determine the magnitude of the total response to structural and accessory proteins, the average number of cells proliferating in response to any of the peptides M, N, ORF3, 6, 7, 8 was obtained as a function of their respective $CD4^+$ or $CD8^+$ T cell population and then expressed as a percentage. The background was then subtracted from the total response for each volunteer.

**Flow cytometry staining.** A MIFlowCyt file (minimum information about a flow cytometry experiment) was created as per Section VI. 4 of 'Guidelines for the use of flow cytometry and cell sorting in immunological studies'[30] and recommended by the International Society for Advancement of Cytometry[31]. The file contains details of antibodies, reagents, instrument settings, gating strategies and controls used for flow cytometry experiments and is provided in the Supplementary Information of this manuscript. PBMC were resuspended in cell staining buffer (Biolegend) in case of proliferation assays or 1×PBS in case of ICS assays and incubated for 20 min with near-infra-red live/dead or aqua fixable stain, respectively (Invitrogen, Carlsbad, CA, USA). Cells from proliferation assays were incubated with fluorochrome-conjugated primary human-specific antibodies for CD3, CD4 and CD8 in cell staining buffer (Biolegend) containing serum for 30 min at 4 °C, washed with cell staining buffer, fixed with 4% paraformaldehyde (PFA, Sigma) and stored at 4 °C in the dark until data acquisition. Cells from ICS assays were fixed with fixation/permeabilization solution (BD Biosciences) for 20 min at 4 °C, washed with permeabilization buffer (BD Biosciences) followed by incubation with fluorochrome-conjugated human-specific antibodies. After washing with permeabilization buffer, the samples were resuspended in 1×PBS and stored at 4 °C in the dark until data acquisition. Data were acquired on an LSRII (BD Biosciences) or MACSquant analyser 10 (Miltenyi) flow cytometer and analysis was performed with FlowJo Version 10.7.1 (BD Biosciences). Specific gating strategies can be found in the Supporting information.

**Lactate measurements.** Supernatants from the proliferation assay were analysed using a published cellular lactate assay[32]. Briefly, colorimetric L-lactate assay kits (Abcam, Cambridge, UK) were used as per the manufacturer's instructions. A standard concentration curve was defined, and the lactate concentration in each day 4 supernatant from the proliferation assay was calculated using a 96-well plate reader.

The lactate proliferation index was calculated on a per-well basis using the following Eq. 1:

$$\text{Proliferation}(\%) = 100 \times (T_{\text{Stim}} - \text{mean}(T_{\text{DMSO}}))/T_{\text{Stim}} \quad (1)$$

where $T_{\text{Stim}}$ is the concentration of lactate for a given well with either PHA or SARS-CoV-2 peptides, and mean ($T_{\text{DMSO}}$) is the average background lactate production from negative control wells.

A significant proliferative response to a given peptide was greater than 0, as determined by Eq. 2:

$$\text{Significance} = \text{mean}(T_{\text{Stim}}) - 3 \times \text{std}(T_{\text{Stim}}) \quad (2)$$

where mean($T_{\text{Stim}}$) is the mean % proliferative response of a specific participant to a stimulus, and std($T_{\text{Stim}}$) is the standard deviation of the participant to a given stimulus.

**Standardised ELISA for detection of spike-specific total IgG in plasma.** Total anti-SARS-CoV-2 spike antibodies were determined using an indirect ELISA[29], which is based on the Krammer assay[33] using a standard curve derived from a pool of SARS-COV-2 convalescent plasma samples on every plate. Standardised EUs were determined from a single dilution of each sample against the standard curve which was plotted using the 4-Parameter logistic model (Gen5 v3.09, BioTek). Each assay plate consisted of samples and controls plated in triplicate, with ten standard points in duplicate and four blank wells.

**SARS-CoV-2 pseudotype micro-neutralisation assay.** Frozen plasma samples were thawed, heat-inactivated at 56 °C for 30 min, and assayed for neutralisation of a lentivirus-based viral particle carrying a luciferase reporter and pseudotyped with full-length SARS-CoV-2 spike (Accession No: YP_009724390.1)[34]. Briefly, neutralising antibody titres were determined by incubating serial two-fold plasma dilutions with ~$10^5$ RLU pseudotyped virus for 2 h before the addition of $10^4$ HEK293T cells transfected with full-length human ACE2 24 h prior. After 72 h incubation at 37 °C, luciferase expression was quantified using BrightGlow (Promega Corp.), readouts were normalised, and −Log(IC50) determined via non-linear regression using GraphPad Prism 8 (GraphPad Software).

**Enzyme-linked immunosorbent assay for coronaviruses.** 229E, NL63, HKU1 and OC43 spike antibody responses were measured using ELISAs. 229E, NL63, HKU1 and OC43 spike antigens were bought from Sino Biological, China. Nunc-Immuno 96-well plates (ThermoFischer Scientific, USA) were coated with 2.0 µg/ml of antigen in PBS buffer and left overnight at 4 °C. Plates were washed with 3× with 0.1% PBS–Tween (PBS/T), then blocked with casein in PBS for 1 h at room temperature (RT). Serum or plasma was diluted in casein–PBS solution at 1:100 dilutions before being added to Nunc-Immuno 96-well plates in triplicate. Plates were incubated for 2 h before being washed with 6× with PBS/T. Secondary antibody rabbit anti-human whole IgG conjugated to alkaline phosphatase (Sigma, USA) was added at a dilution of 1:1000 in casein–PBS solution and incubated for 1 h at RT. After a final wash, plates were developed by adding 4-nitrophenyl phosphate substrate in diethanolamine buffer (Pierce, Loughborough, UK), and optical density OD was read at 405 nm using a BMG Labtech microplate reader. A reference standard comprising of pooled cross-reactive serum and naive serum on each plate served as positive and negative controls, respectively. The positive reference standard was used on each plate to produce a standard curve from which 'relative ELISA units' were derived. Pooled HCoV highly reactive sera were used as a standard for the HCoV spike ELISAs.

**Alignment score analysis.** SARS-CoV-2 sequence was downloaded from GenBank (accession number: NC_045512). All spike, M and NP protein sequences of each of the HCoV species (OC43, HKU1, NL63 and 229E) were downloaded from NCBI using a protein blast. For each protein of each species, all sequences were aligned using MAFFT server (https://mafft.cbrc.jp/alignment/server/) and a consensus sequence was constructed. The clinical sample with the least number of mismatches relative to the consensus sequence was chosen as a representative sequence for each protein of each HCoV species (see Supplementary Table 7, use https://www.ncbi.nlm.nih.gov/protein/ for search). Matlab version R2018b was used to align each SARS-CoV-2 peptide against HCoV proteins. Needleman–Wunsch algorithm as implemented in Matlab (nwalign function) was used to perform a semi-global alignment of each peptide and the alignment score was recorded.

**Statistical analyses.** Statistical analysis was performed with IBM SPSS Statistics 25 and figures were made with GraphPad Prism 8. Chi-square was used to compare the ratio difference between the two groups. After testing for normality using Kolmogorov–Smirnov test, independent-samples t-test or Mann–Whitney U test was employed to compare variables between two groups, and Kruskal–Wallis–ANOVA with Dunn's multiple comparisons test was performed to compare variables between three or more groups with a non-parametric distribution. The correlation was performed via Spearman's rank correlation coefficient. For ICS cytokine expression analyses, data were prepared using PESTEL v2.0 for formatting and baseline subtraction, followed by the export of data to SPICE v6.0 for analysis. Statistical significance was set at $P < 0.05$ and all tests were two-tailed.

**Reporting summary**. Further information on research design is available in the Nature Research Reporting Summary linked to this article.

## Data availability

The SARS-CoV-2 sequence was accessed in the NCBI Gene database under accession code NC_045512. All other data are present in the article and its Supplementary Information files or from the corresponding author upon reasonable request. Source data are provided with this paper.

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

## Acknowledgements

We wish to thank Kathryn Southey and Suki Kenth for administrative support, and our medical student volunteers Julie Dequaire, Rory Fairhead, Shayan Fassih, Thomas Foord, David Kim, Thomas Ritter, Adan Taylor and Rebecca Young for logistical assistance. The views expressed in this article are those of the authors and not necessarily those of the National Health Service (NHS), the National Institutes for Health Research (NIHR) or the Medical Research Council (MRC). This work was supported by the UK Department of Health and Social Care as part of the PITCH (Protective Immunity from T cells to Covid-19 in Health workers) Consortium, the UK Coronavirus Immunology Consortium (UK-CIC), the Huo Family Foundation and the National Core Study: Immunity (NCSi4P programme) 'Optimal cellular assays for SARS-CoV-2 T cell, B cell and innate immunity'. M.A.A. is supported by a Wellcome Trust Sir Henry Dale Fellowship (220171/Z/20/Z). E.B. and P.K. are NIHR Senior Investigators and PK is funded by WT109965MA and NIH (U19 I082360). S.J.D. is supported by an NIHR Global Research Professorship. T.D. is funded by the Medical Research Council (MRC), UK: Chinese Academy of Medical Sciences (CAMS) Innovation Fund for Medical Sciences (CIFMS), China (grant number: 2018-I2M-2-002). D.W.E. is a Robertson Foundation Fellow. A.J.M. was supported by the Medical Research Council [grant MC_PC_19059] and the National Institutes for Health and Oxford Biomedical Research Centre. P.C.M. is funded by a Wellcome intermediate fellowship, ref. 110110/Z/15/Z. The work by A.S. and D.W. was funded by the National Institutes of Health contract Nr. 75N9301900065. D.S. is supported by the NIHR Academic Clinical Fellow programme in Oxford. O.S. is funded by the Wellcome Trust Infection, Immunology, and Translational Medicine programme (Grant: 108869/Z/15/Z). L.T. is supported by the Wellcome Trust (grant number 205228/Z/16/Z). L.T. and P.K. are in the National Institute for Health Research Health Protection Research Unit (NIHR HPRU) in Emerging and Zoonotic Infections (NIHR200907) at the University of Liverpool in partnership with Public Health England (PHE), in collaboration with Liverpool School of Tropical Medicine and the University of Oxford. P.K. is funded by Department of Health and Social Care (DHSC)/UKRI/NIHR COVID-19 Rapid Response Grant (COV19-RECPLA). The views expressed are those of the author(s) and not necessarily those of the NHS, the NIHR, the Department of Health or Public Health England.

## Author contributions

S.J.D., P.K., E.B., J.F., P.G., A.J.M. and H.A.F. conceptualised the project. S.J.D., P.K., E.B., J.F., P.G., H.A.F., A.O., B.K., A.B., J.T.G., M.Pace, E.A., L.L., L.T., M.O., P.T., D.J.N., T.d.S., S.R.J. and M.Pirmohamed designed and supervised T cell experiments. A.J.P., G.S., T.L., C.D., C.T., O.L.S. and N.L. designed and supervised antibody experiments. A.O., A.B., J.T.G., M.Pace, E.A., L.L., M.O., P.T., M.P P.R., M.A., H.B., S.C., P.Z., V.V., C.d.L., T.D., J.G., C.-P.H., W.L., K.J., S.-E.A., A.A., K.A. and H.D.A. performed T cell experiments. T.L., C.D., A.F., O.L.S. and N.L. performed antibody experiments. A.E.B., S.J.D., J.F., D.T.S, K.J., C.P.C., A.J.M., D.W.E., L.S., C.V.A.-C. and P.Z. and the Oxford

Protective T cell Immunology for COVID-19 (OPTIC) Clinical team established the clinical cohorts and collected the clinical samples and data. G.O., T.D., P.C.M., C.T., A.S., D.W., M.A.A., S.C.M., N.R., P.O., R.T. and the Oxford Immunology Network Covid-19 Response T cell Consortium provided critical reagents, technical and intellectual expertise. E.B., S.J.D., J.F., D.T.S. A.O., B.K., A.B., J.T.G., M.Pace, O.L.S., L.T., T.d.S., M.O., P.J.T. and D.J.N. analysed the data. S.J.D., P.K., A.O. B.K., M.Pace and J.T.G., wrote the original draft. S.J.D., P.K., A.O., E.B., J.T., P.G., L.T., T.d.S., A.B., M.Pace, J.T.G., C.D., T.L., G.O., T.D., P.C.M. and P.O. reviewed and edited manuscript and figures.

## Competing interests

D.W.E. declares lecture fees from Gilead. A.S. is listed as an inventor on patent application no. 63/012,902, submitted by La Jolla Institute for Immunology, covers the use of the megapools and peptides thereof for therapeutic and diagnostic purposes. A.S. is a consultant for Gritstone and Flow Pharma and Avalia. The remaining authors declare no competing interests.

## Additional information

[1]Nuffield Department of Clinical Medicine, Peter Medawar Building for Pathogen Research, University of Oxford, Oxford, UK. [2]Nuffield Department of Clinical Medicine, Centre for Tropical Medicine and Global Health, University of Oxford, Oxford, UK. [3]Oxford University Hospitals NHS Foundation Trust, Oxford, UK. [4]Nuffield Department of Clinical Neurosciences, University of Oxford, Oxford, UK. [5]Department of Molecular and Clinical Pharmacology, MRC Centre for Drug Safety Science, University of Liverpool, Liverpool, UK. [6]The Florey Institute for Host-Pathogen Interactions and Department of Infection, Immunity and Cardiovascular Disease, Medical School, University of Sheffield, Sheffield, UK. [7]Translational Gastroenterology Unit, University of Oxford, Oxford, UK. [8]Oxford Vaccine Group, Department of Paediatrics, University of Oxford, Oxford, UK. [9]NIHR Oxford Biomedical Research Centre, University of Oxford, Oxford, UK. [10]MRC Human Immunology Unit, MRC Weatherall Institute of Molecular Medicine, University of Oxford, Oxford, UK. [11]Chinese Academy of Medical Science Oxford Institute (COI), University of Oxford, Oxford, UK. [12]Big Data Institute, Nuffield Department. of Population Health, University of Oxford, Oxford, UK. [13]Jenner Institute, University of Oxford, Oxford, UK. [14]Department of Infection Biology, Faculty of Infectious and Tropical Diseases, London School of Hygiene and Tropical Medicine, London, UK. [15]Department of Physiology, Anatomy, and Genetics, University of Oxford, Oxford, UK. [16]Institute of Cancer and Genomic Sciences, University of Birmingham, Birmingham, UK. [17]Wellcome Centre for Human Genetics, University of Oxford, Oxford, UK. [18]HPRU in Emerging and Zoonotic Infections, Institute of Infection, Veterinary and Ecological Sciences, University of Liverpool, Liverpool, UK. [19]Faculty of Medicine, National Heart and Lung institute, Imperial College, London, UK. [20]Mahidol-Oxford Tropical Medicine Research Unit, Bangkok, Thailand. [21]Nuffield Department. of Clinical Medicine, University of Oxford, Oxford, UK. [22]Center for Infectious Disease and Vaccine Research, La Jolla Institute for Immunology, La Jolla, CA, USA. [23]Department of Medicine, Division of Infectious Diseases and Global Public Health, University of California, Los Angeles, California, USA. [24]Peter Medawar Building for Pathogen Research, Department of Zoology, University of Oxford, Oxford, UK. [25]Peter Medawar Building for Pathogen Research, Department of Paediatrics, University of Oxford, Oxford, UK. [26]Tropical and Infectious Disease Unit, Liverpool University Hospitals NHS Foundation Trust, Member of Liverpool Health Partners, Liverpool, UK. [27]Diabetes Trials unit, Radcliff Department of Medicine, University of Oxford, Oxford, UK. [28]Nuffield Department of Clinical Medicine, University of Oxford, Oxford, UK. [29]Experimental Medicine, Nuffield Department of Clinical Medicine, University of Oxford, Oxford, UK. [30]Oxford University Medical School, University of Oxford, Oxford, UK. [31]These authors contributed equally: Ane Ogbe, Barbara Kronsteiner, Donal T. Skelly, Matthew Pace, Anthony Brown, Emily Adland. [32]These authors jointly supervised this work: Lance Turtle, Paul Klenerman, Philip Goulder, John Frater, Eleanor Barnes, Susanna Dunachie. *Lists of authors and their affiliations appear at the end of the paper. ✉email: paul.klenerman@ndm.ox.ac.uk

## Oxford Immunology Network Covid-19 Response T Cell Consortium

Graham Ogg[3,9,10], Jeremy Chalk[27], Georgina Kerr[28], Prabhjeet Phalora[1], Anna Csala[1], Mathew Jones[1], Nicola Robinson[1], Rachael Brown[1], Claire Hutchings[1], Nicholas Provine[29], Jeremy Ratcliff[1] & Ali Amini[29]

## Oxford Protective T Cell Immunology for COVID-19 (OPTIC) Clinical Team

Susanna Dunachie[1,2,3,9,20,32], Martyna Borak[7], Stavros Dimitriadis[30], Thomas Fordwoh[30], Bryn Horsington[7], Sile Johnson[30], Jordan Morrow[7], Yolanda Warren[7] & Charlie Wells[3]

