## [Peer Review File · Nature Communications]

REVIEWER COMMENTS

Reviewer #1 (Remarks to the Author):

Ogbe and colleagues used a number of SARS-CoV-2 specific T cell assays to differentiate between clinical, subclinical and unexposed individuals. They found T cell responses using IFN ELISPOT and proliferation assay towards M, NP and PRF3 in SARS-CoV-2 exposed individuals but not in pre-pandemic or unexposed samples. Stimulation with S1/S2 overlapping peptides, however, led to proliferation in convalescent and unexposed donors. The authors conclude that the analysis of memory responses towards non-spike proteins versus spike proteins can distinguish SARS-CoV2 exposed versus non-exposed individuals. Understanding T cell immunity in COVID-19 is of importance, although findings on T cell responses towards overlapping peptide pools have been reported by other groups.

Specific comments:

The authors provide the flow cytometry data to show polyfunctionality of SARS-CoV-2 CD4+ and CD8+ T cell responses. Looking at the representative FACS profiles in Supp Fig 4, seven CD4+ T cells and eight CD8+ T cells produced IFN/TNF. The authors then went on and analysed these minimal/background level events for the expression of CD154, CD107a, IFN-g, TNF, IL-2 according to the number of functions: 1-4 functions. How can 7 T cells be so comprehensively analysed into several subsets? I have major concerns with the analysis of these data. As a principle, there should be a threshold for a number of cells (e.g. 10-20 events) which should be no further analysed functionally or phenotypically as the data are questionable.

How many events were gated for the analyses in Supp Fig 5?

Representative FACS plots for Fig 6 and Supp Fig 6 should be shown. How many events were found for seronegative and seropositive controls?

Recent reports revealed dominance of CD4+ T cells over CD8+ T cells in severe COVID-19, while CD8+ T cells appear to dominate in mild COVID-19. Findings in this manuscript should be also analysed to understand CD4 versus CD8 T cell dominance in seropositive and seronegative individuals.

In the Abstract, replace 'doctors' with 'health care workers'.

Reviewer #2 (Remarks to the Author):

In this manuscript, Ogbe et al compare different T cell assays (ELISPOT, ICS and proliferation assays) in well-defined patient cohorts. They claim that ex vivo ELISPOTS can specifically identify COVID-19 convalescent donors, without detecting background in unexposed and pre-pandemic controls. Furthermore, they demonstrate that T-cells from pre-pandemic and unexposed donors proliferate in response to S stimulation, but none of the other antigens (as "proof" for cross-reactivity). COVID-19 convalescent donors responded to almost all viral antigens with rapid proliferation. The main novelty of this manuscript is the discriminatory value of the different assays (T-cell responses in different cohorts have already been described often), the serious weakness is that there is no evidence that responses in pre-pandemic or seronegative donors are really cross-reactive responses induced by seasonal HCoVs.

Major comments:

- The authors often speculate about cross-reactive responses, however provide no real evidence that the responses detected in controls are actually cross-reactive responses. To this end, it is crucial to correlate their "cross-reactive" responses to HCoV-specific antibody or T-cell responses. I find it surprising that S1 and S2 stimulation detects cross-reactive responses, whereas none of the other antigens do. How do the authors explain this, there is definitely conservation in M and N with the seasonal HCoVs? Furthermore, Braun et al (reference 19) did a similar comparison between S1 and S2 stimulation (although they did not use a proliferation assay) and found responses to S2 in controls, not to S1. In Mateus et al (reference 21), cross-reactivity against other antigens than S is also observed? Is the proliferation assay "overestimating" specific T cell responses?
- The last sentence of the discussion is really "overarching", as the authors have not addressed T cells induced by vaccination, and I argue have not provided proof for detection of HCoV-induced T cells.
- Which samples and cohorts are tested in different figures should be clearer. This is crucial information, but not directly visible from all figures. Furthermore, the authors should refrain from making statements on correlation between disease severity and T cell responses. The cohorts contain a lot of potential confounders (age, sex, disease onset time), and there are discrepancies between figure 1, 2 and sup figure 3, where differences are not consistently observed.
- The authors should not compare their data to ELISPOT data from the ChAdOx1 vaccine trial, as the assay is not identical.
- The peak of the T cell responses at day 28-32 is not apparent at all, and the authors mention this both in the results and discussion. What data is the basis for this assumption? I propose this should be removed from the manuscript.
- Have the authors correlated antibody and T-cell responses in these studies at all? Is this relevant, or is there a reason why this obvious correlation was not performed?
- The ICS data seems to be a bit "by itself" and does not contribute to the main message. The fact T cells can be detected by ICS, and express multiple cytokines (I don't like the term "polyfunctional), is not novel. This experiment needs to be implemented in the main message, or considered removing.
- Discussion: although the ELISPOT seems discriminative, the assay also "misses" some asymptomatic positive individuals. The authors should address this in the discussion. Similarly, proliferative responses to M and NP were significantly less in controls, but not absent. The authors should not "overclaim".

Minor comments:

- Introduction: are the majority of infections "asymptomatic"? If true, please reference.
- Introduction: cross-reactivity between SARS-CoV and SARS-CoV-2 is not the same as cross-reactivity between HCoVs and SARS-CoV-2. The conservation is totally different. This needs to be rephrased.
- Introduction: the role of pre-existing cross-reactive T cell responses was also extensively studied in influenza viruses, not only flaviviruses. Please be precise.
- Introduction: initially the authors report 50% cross-reactivity, later they re-emphasize 0-50% cross-reactivity. This is confusing and a contradictory repeat.
- Figure 1: It is not intuitive that every bar in 1C and 1D represents a donor, as the x-axis shows days post onset. Should be clearer. Furthermore, the authors report specific higher responses to certain minipools, but this is not visible in the figure as these are not directly compared. Statistics?
- Figure 1a and Sup Figure 1a do not show identical samples / cohorts. Is this on purpose?
- Figure 1a and Sup Figure 1a: The authors sometimes show ELISPOT data on a log scale, sometimes on a linear scale (correlations). This should be consistent throughout.
- Throughout the authors speculate about effector and central memory T cells, however they have not stained any separate populations (also not in ICS). The basis for this statement is an ex vivo assay versus a proliferation assay, but of course distinguishing these populations with these assays is not black and white. The authors should be careful with statements on phenotype.
- Supplementary figure S3 has an incorrect Y-axis.
- Figure 3: why are S1 and S2 only tested in a limited number of donors?
- When I directly compare figure 6A and 6B with 6C and 6D, there seems to be a difference

between negative donors and pre-pandemic donors. The authors address this in supplemental figure 6 (and state there is no difference). Can the authors explain this discrepancy?
- Figure 7: sometimes 10 and sometimes 8 donors. This discrepancy is not explained.

Response to Nature Communications reviewers' comments

Note: Reviewer's original comments are in black

Responses are in Red (with corresponding extra text or graphs added as screen grabs).

Manuscript title: T cell assays differentiate clinical and subclinical SARS-CoV-2 infections from cross-reactive antiviral responses

REVIEWER COMMENTS

Reviewer #1 (Remarks to the Author):

Ogbe and colleagues used a number of SARS-CoV-2 specific T cell assays to differentiate between clinical, subclinical and unexposed individuals. They found T cell responses using IFN ELISPOT and proliferation assay towards M, NP and PRF3 in SARS-CoV-2 exposed individuals but not in pre-pandemic or unexposed samples. Stimulation with S1/S2 overlapping peptides, however, led to proliferation in convalescent and unexposed donors. The authors conclude that the analysis of memory responses towards non-spike proteins versus spike proteins can distinguish SARS-CoV2 exposed versus non-exposed individuals. Understanding T cell immunity in COVID-19 is of importance, although findings on T cell responses towards overlapping peptide pools have been reported by other groups.

Specific comments:

The authors provide the flow cytometry data to show polyfunctionality of SARS-CoV-2 CD4+ and CD8+ T cell responses. Looking at the representative FACS profiles in Supp Fig 4, seven CD4+ T cells and eight CD8+ T cells produced IFN/TNF. The authors then went on and analysed these minimal/background level events for the expression of CD154, CD107a, IFN-g, TNF, IL-2 according to the number of functions: 1-4 functions. How can 7 T cells be so comprehensively analysed into several subsets? I have major concerns with the analysis of these data. As a principle, there should be a threshold for a number of cells (e.g. 10-20 events) which should be no further analysed functionally or phenotypically as the data are questionable.

Reply: Reviewer 1 makes a reasonable criticism of the data—based on the representative plot we showed, he/she assumed we did not obtain enough cells for polyfunctional analysis. The reviewer was fine with the ICS summary data, just not the polyfunctional analysis. The representative figure was chosen for strength of signal rather than the amount of positive cells and we have now replaced it with a new representative figure to better present our data.. We have re-examined our data to exclude any patients that did not have enough cells to meet the reviewer's criteria. We then reperformed the polyfunctional analysis. Because of the overall low magnitude of the ICS response, we did end up excluding more subjects and had to remove polyfunctional CD8 analysis, as we didn't have sufficient patients who met the reviewer's criteria. However, our CD4 data had enough donors for analysis (n=9 for M response and n=4 for NP response, see below figure) and the data was very similar to our original analysis. None of our conclusions changed.

Figure 4: ICS responses in CD4+ and CD8+ T cells for M and NP pools in ELISpot positive individuals. ICS was performed on individuals with convalescent mild cases and a positive ELISpot for the indicated peptides. PBMC were stimulated with 2ug/mL peptide for 6 hours. Expression levels of IFN- γ , IL-2 and TNF- α in CD4+and CD8+T cells using M pools are shown in a), n=31. Bars represent median +/- IQR. Statistics were performed using Wilcoxon matched-pairs signed rank test between each cytokine in CD4+vs CD8+T cells. Where sufficient cells were available for analysis, Boolean gates were set and polyfunctionality was examined in CD4+T cells (b) using SPICE (n=9) where CD107a, IFN- γ , IL-2, TNF- α , and CD154 were examined. Error bars represent SEM for polyfunctionality figures. Expression levels of cytokines using NP pools are shown in (d) (n=41) with polyfunctionality analysis for CD4+T cells (n=4) shown in (d).

We did the same thing for the polyfunctional analysis shown in Supp Fig 5 with similar results. Again, only the CD4 cells had enough cells for polyfunctional analysis with sufficient n (S1 n=7, S2 n=3, M n=10, NP n=4). Again, there was no significant change to the data and our conclusions have not changed (see figure below).

Supplementary Fig 5: ICS responses in CD4+ and CD8+ T cells for Spike Pools. ICS was performed as in Figure 4 on n=26 individuals PCR+ for SARS-CoV-2. Expression levels of IFN- γ , IL-2 and TNF- α in CD4+ and CD8+ T cells are shown for the peptide pools M in (a), NP in (c), S1 in (e), and S2 in (g). Bars represent median +/- IQR. Statistics were performed using Wilcoxon matched-pairs signed rank test between each cytokine in CD4+ vs CD8+ T cells. Polyfunctionality was then assessed as in Figure 4. Polyfunctionality for M pools is shown for CD4+ T cells in (b) (n=10). D shows polyfunctionality for NP pools in CD4+ T cells (n=4). Polyfunctional responses with S1 pools (n=7) and S2 pools (n=3) are shown for CD4+ T cells in (f) and (h) respectively.

Supplemental Figure 4: Representative ICS plots. The gating strategy is shown in (a). In (b), PBMC were stimulated with 2ug/mL of the indicated peptide pool or DMSO control and representative plots for gated CD4+T cells is shown.

Overall, we think we can address this point by careful reanalysis of the existing dataset, without changing the overall conclusions – indeed strengthening them. In this revision we are also addressing the point made by second reviewer to better integrate the ICS data (rather than necessarily remove it). In sum, we feel that the ICS in its new reanalysed form can nicely support the other data – considering we are comparing different assay forms this is still relevant - and contribute to the conclusions of the manuscript

How many events were gated for the analyses in Supp Fig 5?

Reply: This has been addressed above.

Representative FACS plots for Fig 6 and Supp Fig 6 should be shown. How many events were found for seronegative and seropositive controls?

Reply: We have shown representative proliferation in Supp Figure 2, which show positive and negative controls. We can clarify the relative cell numbers in the spike responses in the different patient groups. Overall the assays were performed in an identical way and gave results which are not statistically significant but we can include more data to show that we acquired similar cell numbers to achieve these results

Recent reports revealed dominance of CD4+ T cells over CD8+ T cells in severe COVID-19, while CD8+ T cells appear to dominate in mild COVID-19. Findings in this manuscript should be also analysed to understand CD4 versus CD8 T cell dominance in seropositive and seronegative individuals.

Reply: We do compare CD4 vs CD8 responses in Fig 4. in a peptide, cytokine-based way. We have clarified this point in the text and the figures in the revised manuscript.

6. In the Abstract, replace 'doctors' with 'health care workers'.

Reply: This has been done as suggested by the reviewer. See line 73.

Reviewer #2 (Remarks to the Author):

In this manuscript, Ogbe et al compare different T cell assays (ELISPOT, ICS and proliferation assays) in well-defined patient cohorts. They claim that ex vivo ELISPOTS can specifically identify COVID-19 convalescent donors, without detecting background in unexposed and pre-pandemic controls. Furthermore, they demonstrate that T-cells from pre-pandemic and unexposed donors proliferate in response to S stimulation, but none of the other antigens (as “proof” for cross-reactivity). COVID-19 convalescent donors responded to almost all viral antigens with rapid proliferation. The main novelty of this manuscript is the discriminatory value of the different assays (T-cell responses in different cohorts have already been described often), the serious weakness is that there is no evidence that responses in pre-pandemic or seronegative donors are really cross-reactive responses induced by seasonal HCoVs.

Major comments:

7. - The authors often speculate about cross-reactive responses, however provide no real evidence that the responses detected in controls are actually cross-reactive responses. To this end, it is crucial to correlate their “cross-reactive” responses to HCoV-specific antibody or T-cell responses. I find it surprising that S1 and S2 stimulation detects cross-reactive responses, whereas none of the other antigens do. How do the authors explain this, there is definitely conservation in M and N with the seasonal HCoVs? Furthermore, Braun et al (reference 19) did a similar comparison between S1 and S2 stimulation (although they did not use a proliferation assay) and found responses to S2 in controls, not to S1. In Mateus et al (reference 21), cross-reactivity against other antigens than S is also observed? Is the proliferation assay “overestimating” specific T cell responses?

The reviewer is correct that we had not in this manuscript examined the origin of the responses we observe in the pre-pandemic or negative control samples. However, this issue has been now addressed. Overall, we think our data are quite consistent with those studies mentioned (including data from our co-authors from the La Jolla group) and where cross-reactivity was formally assessed at a peptide-specific level. To improve the manuscript here in response to this we have used three approaches

- We tested as suggested our samples from our control cohort (and some of the seropositive group) for serologic reactivity to the 4 circulating coronaviruses (OC43, HKU1, NL63 and 229E). We found strong responsiveness to all viruses tested. This does not prove T cell reactivity is driven by prior exposure to these viruses but it shows substantial exposure to these viruses with appropriate immunologic priming in this group. It is also consistent with the published literature in the Mateus reference. These data are shown in Supplementary Figure 7 and discussed in the text.
- We next looked at the antigen conservation of S, M and NP and calculated the level of relatedness of these antigens at the level of peptides. Two features emerged from this. Firstly the median alignment score (indicating conservation of peptides) was higher in S than M or N. This result is highly significant (p-values: S vs. N < 2.2×10^{-16} , S vs. M < 2.2×10^{-16}). Secondly, the number of peptides with the highest high alignment scores (>18.5, top 5%) was dominated by those from S. Thus we conclude the slightly higher degree of conservation at the level of peptides – coupled with the larger size of S, give responses to this protein a significant advantage. (Conservation between different coronaviruses extended over multiple reinfections in the past may serve to accentuate this advantage). These independent data are entirely consistent with the paper by Mateus and colleagues which showed a marked dominance of S specific responses in Covid-negative controls. This is shown in Supplementary Figure 8a and discussed in the text
- Next we addressed the question about cross-reactivity posed by the reviewer by reanalysing data from Mateus et al together with our co-authors on that paper. We looked at the 31 S peptides where cross-reactive epitopes in circulating strains were assessed and the 30 peptides in Non-S antigens (Structural and ORF1 derived proteins). We plotted the numbers of individuals responding to these cross-reactive/conserved peptides for S and non-S peptides and showed a marked skewing towards responses to the S antigen. We also plotted the distribution of these in S1 and S2 and found reactivity in both regions of S. These data are shown in Supplementary Figure 8b and c and discussed in the same text as above.

From all this we conclude that the group are extensively exposed, that cross-reactivity can occur across the coronavirus genome, but that it is weighted towards S and our assays – combined with existing data showing functional cross-reactivity in responses to the same antigens - reflect that. Cross-reactive responses to M and N as well as other proteins can be generated but much less frequently. Having said all this – for any given individual, we do not know the past history of these responses so we have edited the text carefully to take that into consideration, and discussed this issue in the conclusions.

8. The last sentence of the discussion is really “overarching”, as the authors have not addressed T cells induced by vaccination, and I argue have not provided proof for detection of HCoV-induced T cells.

Reply: we have now amended the text as suggested **See line 485 – 487.**

“These two assays, in combination with the panel of antigens can now allow us to address critical questions about dissecting the role of T cells – induced by SARS-CoV-2, and possibly HCoVs and other antigens - in immune protection in the future.”

9. Which samples and cohorts are tested in different figures should be clearer. This is crucial information, but not directly visible from all figures.

Reply: We have adjusted the description of samples used for each figure to improve the clarity of the figure legends, and provided the source data.

10. Furthermore, the authors should refrain from making statements on correlation between disease severity and T cell responses. The cohorts contain a lot of potential confounders (age, sex, disease onset time), and there are discrepancies between figure 1, 2 and sup figure 3, where differences are not consistently observed.

Reply: We have removed the analysis of disease severity and T cell responses in order to focus on this interesting question in more depth a subsequent manuscript where we will be modelling over time with potential confounders.

11. The authors should not compare their data to ELISPOT data from the ChAdOx1 vaccine trial, as the assay is not identical.

Reply: We have now amended that part of the result section removing any comparison of our data to the SARS-CoV-2 ChAdOx1 vaccine trial.

12. The peak of the T cell responses at day 28-32 is not apparent at all, and the authors mention this both in the results and discussion. What data is the basis for this assumption? I propose this should be removed from the manuscript.

Reply: Thank you, we have removed this from the manuscript and will address this with our timecourse data in a future manuscript. Line 167 now reads “Combined, there was variation in the breadth and magnitude of SARS-CoV-2-specific responses (**Figure S1b**), and longitudinal follow-up studies underway will define the dynamics of the T cell response over time.”

13. Have the authors correlated antibody and T-cell responses in these studies at all? Is this relevant, or is there a reason why this obvious correlation was not performed?

Reply Attn: Thank you, we have added this analysis to the manuscript at line 172: “**Correlation analysis between the IFN- γ responses to spike peptide pools measured by ELISpot and anti-spike IgG measured by ELISA showed a significant positive correlation ($r = 0.4587$; $p < 0.0001$) (Figure 2a)**”. No correlation was found for anti-spike proliferation responses by CTV assays and anti-spike IgG ELISA.

14. The ICS data seems to be a bit “by itself” and does not contribute to the main message. The fact T cells can be detected by ICS, and express multiple cytokines (I don’t like the term “polyfunctional), is not novel. This experiment needs to be implemented in the main message, or considered removing.

Reply: This point we have considered in relation to the comments from Reviewer 1. We felt on balance the inclusion of the ICS data helped add some depth to the ex vivo cytokine findings from

ELISpot. We have improved the integration of these data to make them useful. We have also replayed the word polyfunctional with “expressed multiple cytokines” (line 264).

15. Discussion: although the ELISPOT seems discriminative, the assay also “misses” some asymptomatic positive individuals. The authors should address this in the discussion. Similarly, proliferative responses to M and NP were significantly less in controls, but not absent. The authors should not “overclaim”.

Reply: We wish to be conservative in our interpretation of our findings and we have softened our assertions about the ELISpot assay, including changing “identify” to “characterise” in Line 403 and adding to Line 408 “However, this ELISpot assay did not detect SARS-CoV-2 specific T cell responses in all subjects with recent PCR-confirmed infection”.

Minor comments:

16. Introduction: are the majority of infections “asymptomatic”? If true, please reference.

Reply: The second sentence of the Introduction (Line 84) has been modified to read “While the majority of SARS-CoV-2 infections are either asymptomatic or result in mild disease, some individuals develop severe respiratory symptoms which may result in hospital admission and death leading to high global mortality”.

17. Introduction: cross-reactivity between SARS-CoV and SARS-CoV-2 is not the same as cross-reactivity between HCoV and SARS-CoV-2. The conservation is totally different. This needs to be rephrased.

Reply: we have rephrased this statement for clarity. See line 109 – 113.

18. Introduction: the role of pre-existing cross-reactive T cell responses was also extensively studied in influenza viruses, not only flaviviruses. Please be precise.

Reply: We have added reference to influenza at Line 121.

19. Introduction: initially the authors report 50% cross-reactivity, later they re-emphasize 0-50% cross-reactivity. This is confusing and a contradictory repeat.

Reply: We meant the literature reports divergent data of up to 50% cross reactivity while one particular study show 50% cross reactivity in pre-pandemic samples specifically. We have now rephrased this for clarity. See line 128.

20. Figure 1: It is not intuitive that every bar in 1C and 1D represents a donor, as the x-axis shows days post onset. Should be clearer. Furthermore, the authors report specific higher responses to certain minipools, but this is not visible in the figure as these are not directly compared. Statistics?

Reply: We have amended the X axis labels and figure legends to be clearer that each bar represents an individual donor. We have amended the comment on higher responses in certain minipools to indicate they are examples (line 163). “IFN- γ responses to spike (S) pools were seen in PBMC from 34/75 (45%) of convalescent subjects tested (Figure 1c) with high and frequent responses to some individual minipools including P2 (up to 313 SFC/10⁶ PBMC) and P8 (up to 353 SFC/10⁶ PBMC)”.

21. Figure 1a and Sup Figure 1a do not show identical samples / cohorts. Is this on purpose?

Reply: Assays were performed according to the availability of samples and reagents in the early pandemic phase. We have adjusted the description of samples used for each figure to improve the clarity of the figure legends, and provide the source data.

22. Figure 1a and Sup Figure 1a: The authors sometimes show ELISPOT data on a log scale, sometimes on a linear scale (correlations). This should be consistent throughout.

Reply: All figures now use the linear scale – this has resulted in adjustment of Figure 1b and Supplementary Figures 1g and 1h

23. Throughout the authors speculate about effector and central memory T cells, however they have not stained any separate populations (also not in ICS). The basis for this statement is an ex vivo assay versus a proliferation assay, but of course distinguishing these populations with these assays is not black and white. The authors should be careful with statements on phenotype.

Reply: We have adjusted the text to make the distinction more clearly related to the assays performed.

24. Supplementary figure S3 has an incorrect Y-axis.

Reply: This plot has now been amended to show the right title for the Y axis.

25. Figure 3: why are S1 and S2 only tested in a limited number of donors?

Reply: We did not run the proliferation assay in all PCR+ donors because there were limitations in sample availability. Also, as at the time the study was conducted, we had challenges with availability and supply of peptide pools with Spike peptide pools being the most critical. As these assays were run on fresh samples at the peak of the pandemic, we have included the data we collected. This has been explained in the text.

26. When I directly compare figure 6A and 6B with 6C and 6D, there seems to be a difference between negative donors and pre-pandemic donors. The authors address this in supplemental figure 6 (and state there is no difference). Can the authors explain this discrepancy?

Reply: It is correct that a few individuals make low level responses to other SARS-CoV-2 non –S proteins in the control groups especially in the pre-pandemic controls as discussed above. We believe this is reflective of what can be observed in a randomly selected group of people and so, is truly representative of potentially cross-reactive immunity in the general population. However, comparing the two control groups, these responses in individuals are not significantly different with the exception of ORF3 with very modest significance ($p = 0.04$). We have also amended the plots for clarity and to avoid confusion.

27. Figure 7: sometimes 10 and sometimes 8 donors. This discrepancy is not explained.

Reply: As with point 25 above, we were limited by sample/cell availability and so could not perform the all assays on all subjects.

REVIEWERS' COMMENTS

Reviewer #1 (Remarks to the Author):

The authors greatly improved the manuscript.

Reviewer #2 (Remarks to the Author):

The authors have addressed my comments to satisfaction (with the exception of the cytokine data). Line 220-257 are difficult to interpret, as pointed out by the first reviewer rely on limited events in flow, and limited numbers of donors with variable background.

Furthermore, the focus of this manuscript is on T-cell assays to differentiate between clinical and subclinical infections and cross-reactive responses. This data is now compiled into figure 5 and 6 (comparison between ELISPOT and proliferation), and I find a lot of the characterisation data before (including the ICS) not relevant for the main message per se.

I have few other remaining minor comments:

- line 100: "seronegative infection" is not a term I am familiar with. Rephrase.
- line 145-148: clarify that ELISPOT is used here
- line 169: was Spearman used to calculate R, this is indicated for 2b and 2c, but not 2a
- line 226: I argue that ex vivo ELISPOT and ICS also detect responding CM and EMRA cells. The difference is not so black and white as the authors present here.

Response to Nature Communications reviewers' comments

Note: Reviewer's original comments are in black

Responses are in Red (with corresponding extra text or graphs added as screen grabs).

Manuscript title: T cell assays differentiate clinical and subclinical SARS-CoV-2 infections from cross-reactive antiviral responses

REVIEWER COMMENTS

REVIEWERS' COMMENTS

Reviewer #1 (Remarks to the Author):

The authors greatly improved the manuscript.

Thank you for the review.

Reviewer #2 (Remarks to the Author):

The authors have addressed my comments to satisfaction (with the exception of the cytokine data). Line 220-257 are difficult to interpret, as pointed out by the first reviewer rely on limited events in flow, and limited numbers of donors with variable background.

In response to Reviewer 1 comments on our previous resubmission, we re-examined our data to exclude any patients that did not have enough cells to meet the reviewer's criteria and reperformed the polyfunctional analysis. Our CD4 data had enough donors for analysis (n=9 for M response and n=4 for NP response, and we believe this is now a robust data set which satisfied Reviewer 1's prior concerns.

Furthermore, the focus of this manuscript is on T-cell assays to differentiate between clinical and subclinical infections and cross-reactive responses. This data is now compiled into figure 5 and 6 (comparison between ELISPOT and proliferation), and I find a lot of the characterisation data before (including the ICS) not relevant for the main message per se.

Thank you. We agree that the main message is use of these T cell assays to differentiate between infection and cross-reactive infections, but feel an important part of this package is the demonstration of the ELISPOT, ICS, cellular lactate and proliferation assays in well-defined convalescent cohorts first. In particular, our experience of presenting this data to multiple groups (including UK-CIC and the WHO working group on COVID-19 assays) is that people in our field always ask to see ICS data on cytokine functionality.

I have few other remaining minor comments:

- line 100: "seronegative infection" is not a term I am familiar with. Rephrase.

We agree that this is not the clearest of terms and have revised this to "infection without seroconversion" (line 168)

- line 145-148: clarify that ELISPOT is used here

The subheading of this section now reads "Strong and broad IFN- γ ELISpot responses in convalescence" (Line 206)

- line 169: was Spearman used to calculate R, this is indicated for 2b and 2c, but not 2a

Yes, this is Spearman, and Fig 2a has now been updated to provide the same information as Figs 2b and 2c

- line 226: I argue that ex vivo ELISPOT and ICS also detect responding CM and EMRA cells. The difference is not so black and white as the authors present here.

We have modified the subheading for this section to “Sensitive proliferation assays demonstrate memory responses” in order to tone down any impression that the proliferative response is the sole measurement of central memory.